# Translation and natural selection of micropeptides from long non-canonical RNAs

Pedro Patraquim [1], Emile G. Magny[1], José I. Pueyo [2], Ana Isabel Platero [1] & Juan Pablo Couso [1] ✉

Long noncoding RNAs (lncRNAs) are transcripts longer than 200 nucleotides but lacking canonical coding sequences. Apparently unable to produce peptides, lncRNA function seems to rely only on RNA expression, sequence and structure. Here, we exhaustively detect in-vivo translation of small open reading frames (small ORFs) within lncRNAs using Ribosomal profiling during *Drosophila melanogaster* embryogenesis. We show that around 30% of lncRNAs contain small ORFs engaged by ribosomes, leading to regulated translation of 100 to 300 micropeptides. We identify lncRNA features that favour translation, such as cistronicity, Kozak sequences, and conservation. For the latter, we develop a bioinformatics pipeline to detect small ORF homologues, and reveal evidence of natural selection favouring the conservation of micropeptide sequence and function across evolution. Our results expand the repertoire of lncRNA biochemical functions, and suggest that lncRNAs give rise to novel coding genes throughout evolution. Since most lncRNAs contain small ORFs with as yet unknown translation potential, we propose to rename them "long non-canonical RNAs".

Small, or short, open reading frames (smORFs) are DNA and RNA sequences that could be translated into proteins of less than 100 amino-acids. Hundreds of thousands of smORF sequences are found in eukaryotic genomes[1,2], and thousands can be mapped to transcripts, often to putative non-coding RNAs[3,4], thus challenging our understanding of our genomes' coding potential. smORFs have been deemed non-coding on the basis of their high numbers, small size, and absence of experimental functional evidence, but there is a growing realisation that hundreds, if not thousands of smORFs are translated[3,5–7] and that smORF peptides can have essential functions and be conserved across metazoans[8–10]. However, the full repertoire of smORF functional peptides is not known, nor are the genomic and evolutionary roles of smORF sequences. Metazoan smORFs can be classified into different classes according to their genomic features and translation levels, and these classes might have different molecular functions[11]. In metazoan genomes, (a) hundreds of annotated short coding DNA Sequences (sCDSs) appear mostly in monocistronic mRNAs, robustly translated into peptides about 80 AA long with a

propensity to associate with membranes and regulate canonical (>100 AA) proteins with developmental[8,12] or physiological[9,13] roles; (b) thousands of upstream ORFs (uORFs) are located in the 5′ leaders of canonical mRNAs, not only regulating the translation of the canonical protein located downstream, but also producing short peptides (-25 AA) that can either interact with it[14], or function independently[15,16]; finally, (c) long non-coding RNAs (lncRNAs) contain tens of thousands of smORFs averaging 20 codons in length (lncORFs). lncRNAs are longer than 200 bp but devoid of canonical annotated ORFs, and thus assumed to have non-coding functions. Indeed, several lncRNAs have functions mediated solely by their RNA sequence and structure, from chromatin factors to mRNA translation regulators throught the production of micro-RNAs[17]. However, hundreds of lncRNAs (each containing an average of 20 lncORFs) are polyadenylated and found in the cytoplasm, and can associate with ribosomes, indicating their potential for translation and production of micropeptides[18]. The production of functional micropeptides has been demonstrated in some cases reviewed in refs. 19–22 but it remains unclear to what extent

[1]Centro Andaluz de Biología del Desarrollo, CSIC-UPO, Seville 41089, Spain. [2]Brighton and Sussex Medical School, Falmer BN1 9QG, UK.
✉e-mail: jpcou@upo.es

translation into micropeptides is a general aspect of lncRNA biology, and further, how many of these peptides have biological functions.

smORFs provide an exciting overlap with another nascent field: de novo gene evolution. For a long time, the existence of tens, if not hundreds of species-specific ("orphan") genes has been noted[23–25]. It is as if these genes have appeared not from mutation or duplication of existing genes, but de novo from previously non-coding sequences. However, the de novo gene concept has been beset with problems and controversies around their identification[26–28] where, in effect, absence of evidence—homologies—was taken as evidence of absence. Further, there is currently no consensus on the mechanisms involved in de novo gene creation. It has been suggested that de novo genes arise from proto-genes, sequences with features somehow intermediate between inert DNA and full coding, canonical proteins[29]. Although neither these intermediate features, nor the mechanisms involved in the transition to a de novo gene have been proven, it has also been proposed that lncRNAs and smORFs can provide examples of de novo genes[4,5,11,30], and that the acquisition of translation is a crucial event in the process[11].

Given its ease of use, repertoire of closely-related species with sequenced genomes, and similarity of its smORFs with those of vertebrates[11], *Drosophila melanogaster* provides an ideal model to address the related questions of smORF and lncRNA translation, evolution and function. Previous studies identified ribosomal binding in particular *Drosophila melanogaster* contexts (cell cultures and early embryos)[3,31,32], but did not conclusively prove the translation of lncRNAs. Here we use ribosomal profiling with unprecedented depth and resolution across embryonic development to obtain the characteristic tri-nucleotide periodicity of mapped ribosome protected footprints (RPFs) that proves translation[33,34]. We show that ribosomal binding concentrates to particular lncRNAs and produces either micropeptide translation, or an intermediate 'ribo-bound-only' state, as observed recently in uORFs[5]. Further, we identify molecular features that not only support the observed lncRNA translation, but also the mechanism involved. Finally, our bioinformatics pipeline, GENOR, reveals a striking correlation between lncORF translation and evolution, with translated lncORFs displaying natural selection and older phylogenetic origin than non-translated ones. Taken together our findings are compatible with the possibility that lncRNAs acquire protein-coding function as their lncORFs evolve from an intermediate ribo-bound-only state to a fully translated one.

## Results

### Expanded depth and framing resolution leads to the detection of bona fide in vivo lncORF translation

We identified 16,334 lncORFs mapping to the 2545 genes currently annotated as lncRNAs in *Drosophila melanogaster*[35]. To determine their translation status, we used two biological replicas (T and B) of Poly-Ribo-Seq[5] and mRNA-seq from three 8-h intervals covering the whole of *Drosophila melanogaster* embryogenesis: *Early* (0–8 h), *Mid* (8–16 h) and *Late* (16–24 h). In total, we amassed ca. 2 billion Riboseq and 604 million RNAseq reads, for a total of ca. 570 million genome-mapped Riboseq reads (Supplementary Table 1, methods), that reveal a highly significant correlation between replicates (Fig. 1a, Supplementary Fig. 1e). We followed our recently published smORF-suited pipeline[5], considering a lncORF translated in a given stage if it fulfilled three conditions: (a) transcription (RNAseq signal RPKM$^{RNA}$ > 1 in either replica); (b) ribosome binding (Riboseq signal RPKM$^{FP}$ > 1 in both replicas); and (c) framing in ORF-aligned 32nt RPFs (tri-nucleotide periodicity passing a binomial test $p < 0.01$ in either replica, (Supplementary Fig. 1a). We detected 124 translated lncORFs within the 866 lncRNAs transcribed during *Drosophila melanogaster* embryogenesis. This number is in contrast to the 1258 translated uORFs found using a similar strategy[5], and can be explained by the low RPKM$^{RNA}$ and RPKM$^{FP}$ signal in lncORFs when compared to other ORF classes (Fig. 1b).

A marked improvement came from expanding our analysis parameters. *Drosophila melanogaster* RiboSeq samples show significant RPF size dispersion (26–36 nt, yet consistent with RPF sizes in other metazoans[36,37]) and variable framing patterns per RPF length[3,5,31] (Fig. 1c, d), whose experimental or biological bases remain unclear. Interestingly, our metagene analysis of framing in canonical transcripts revealed that RPFs of different lengths show consistent tri-nucleotide periodicity, yet in different frames (Fig. 1c, d, Supplementary Fig. 1a). Further, RPFs of different sizes show framing in the same or adjacent codons, consistent with originating from the same translated ORF but resulting from variable ribonuclease protection (Fig. 1a). For example, 13,235 (84%) canonical ORFs with framing in 30 and 32 nt RPFs in Late stage Replica B correspond to the same ORF. These data indicate that statistically significant framing patterns in different RPFs reveal the same instances of ORF translation (Fig. 1c, d). lncORF RPFs presented a size dispersion like canonical RPFs (concentrated between 26 and 36 nt). We then ascertained whether non-32 nt. RPF framing in lncORFs also represents bona fide translation signal, by performing a global comparison of framing across canonical ORFs and lncORFs, and correlating the number of reads per frame per RPF length across the two ORF classes. Global framing patterns are very highly correlated between lncORFs and canonical ORFs of the same replica (*rho* between 0.88 and 1 across replicas, see Supplementary Fig. 1b) indicating that reads mapping to lncORFs are as indicative of translation as those of canonical genes.

Thus, we expanded our framing detection to all 26–36 nt RPF sizes, expecting that this increase in mapped reads would improve the number and reliability of framing events in the lowly expressed lncRNA transcripts (Fig. 1b). We found that 292 lncORFs now show translation signal (in any RPF size). Given this 235% increase in translation detections, we took advantage of our experimental replication, measuring framing independently in each T and B replica. 107 lncORFs appear robustly translated (i.e. show framing in both replicas of a given stage) while 185 show limited translation (i.e. framing in only one replica) (Fig. 1e, see also Supplementary Data 1). Comparing our RNA-Seq and Ribo-Seq datasets, we observe 2832 lncORFs that are transcribed, polyadenylated and detected in the cytoplasm in one or more stages, but never showing reproducible ribosomal association, and which we call 'transcribed-only' (Fig. 1e). Interestingly, we also find a group of 331 lncORFs that display reproducible ribosome binding (RPKM$^{FP}$ > 1 in both replicas of a given stage) but no evidence of framing in any sample (Fig. 1e). These lncORFs are comparable to the previously described 'ribo-bound-only' uORFs[5], and could associate with ribosomes for reasons other than productive translation.

Since RPFs of different sizes likely arise from different ribonuclease protection events, and thus from independent molecular events of ribosome-ORF binding, detection of framing in distinct RPF lengths further reinforces translation detection (Fig. 1d, e). We found that robustly translated lncORFs are supported by a minimum of 20 framed reads and between 4 and 85 independent translation events (RPFs showing framing and that are either of different-sizes, or come from different replicas or stages) (Fig. 1d, e, Supplementary Fig. 1c). Interestingly a number of transcripts from characterized lncRNAs with putative non-coding functions (e.g. *IBIN, sphinx, rox2, Hsr-omega, bxd*) show one or more robustly translated lncORFs (Fig. 1f).

To corroborate lncORF translation independently we generated FLAG-tagged versions of lncORFs and quantified their expression in S2 cell cultures. lncORFs with framing corroborated in multiple RPF sizes display strong and reproducible FLAG signal, whereas ribo-bound-only or transcribed-only lncORFs do not (Fig.1f, Supplementary Fig. 1d), even if located within the same RNA transcript (see CR30055 later). Importantly both our Poly-Riboseq and tagging data corroborate the translation of lncRNAs with proposed non-coding functions, such as *IBIN*[38] (Fig. 1g, Supplementary Fig. 1d).

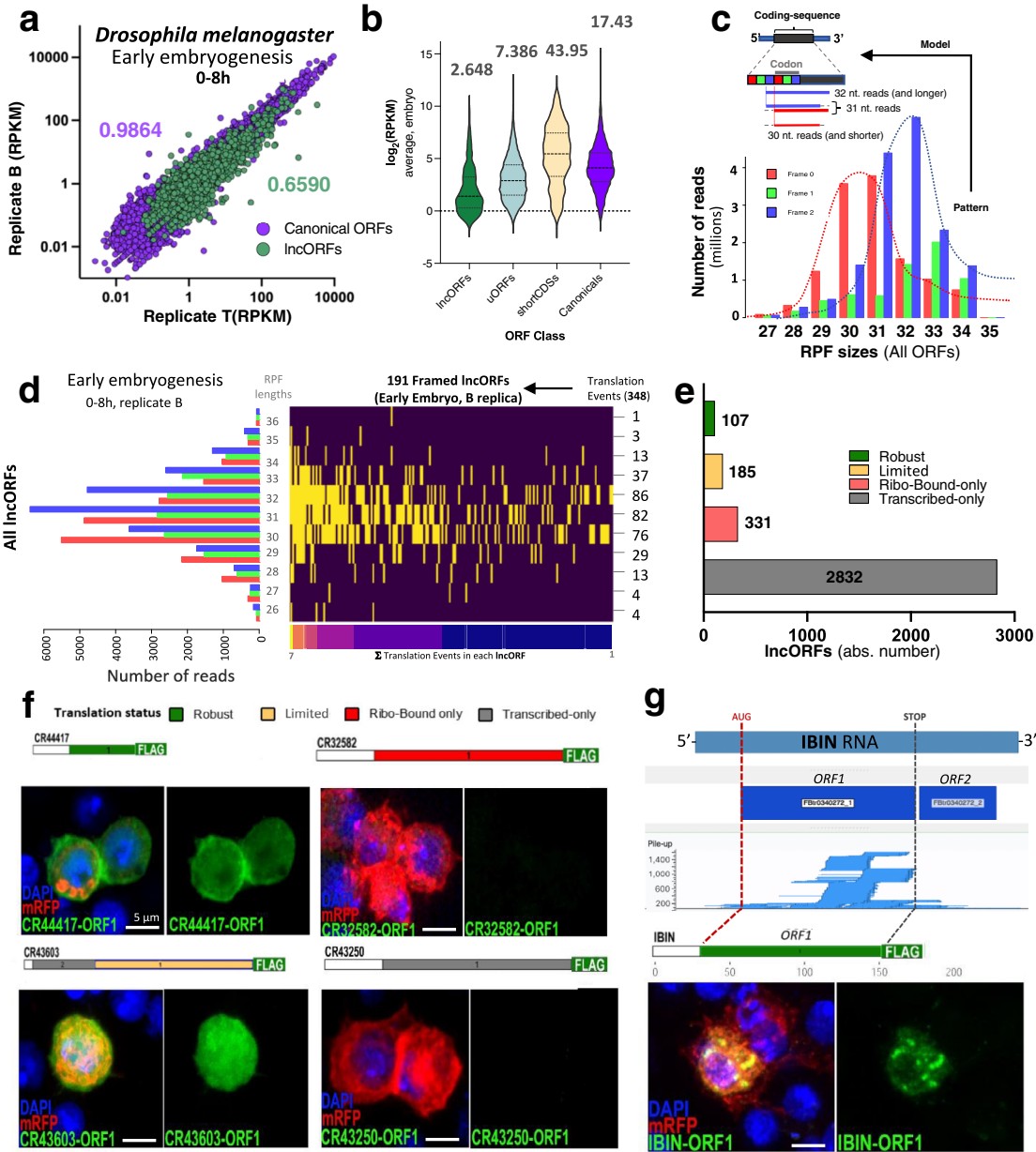

**Fig. 1 | Detecting lncORF translation. a** Correlation plot and Spearman's correlations for lncORF (green) and canonical ORFs (purple) RPKM$^{FP}$ values across replicas B (*y* axis) and T (*x* axis) of early embryogenesis. **b** Average ribosomal binding (RPKM$^{FP}$) across embryogenesis for different types of coding sequences. Median value of the distributions shown on top. **c** Top, framing deconvolution model for *Drosophila melanogaster* Poly-Ribo-Seq. 31-nt long RPFs contain a mix of reads mapping to adjacent frames 0 and 2, reflecting a 1-ribonucleotide loss in either 5′ or 3′-positions, respectively, when compared to 32-nt reads (or longer), which mostly map to frame 2 (blue). 30 nt reads (and shorter) map predominantly to frame 0, explained by a 1-nt. loss in both 5′ and 3′-positions in this population of reads. Bottom, canonical ORF framing across ribosomal footprint lengths 26–36 nt. frame 0 (red), frame 1 (green) and frame 2 (blue). red dotted curve: frame 1 over-representation across shorter RPF length; blue dotted curve: frame 2 over-representation across longer RPF lengths. **d** Redundancy in lncORF translation signal. Left plot: Number of reads per frame for all RPF lengths (26–36 nt) mapping to lncORFs in one biological Poly-Ribo-Seq sample (0–8 h, Replicate B). Heatmap: detected framing events (yellow) per RPF length (y axis) per lncORF (x axis), sorted from higher (7) to lower (1) number of detections; 348 framing events support the translation of 191 lncORFs in this sample. Total number of events per RPF length is indicated on the right. **e** Number of lncORFs according to their translation signal. **f** FLAG-tagged lncORFs with translation signal are translated in S2 cells (left) whereas lncORFs lacking translation signal are not (right). Diagrams represent each construct used; 5′UTRs appear white, and lncORFs as colour-coded according to their Riboseq translation status (top). **g** The immunity-related lncRNA IBIN is robustly translated. ORF1 within the IBIN transcript shows accumulation of RPF reads, but not ORF2. Expression of FLAG-tagged IBIN ORF1 in S2 cells (see **f**) confirms its translated status. Source data are provided as a Source data file.

The *Drosophila* genome annotation uses different standard gene prediction algorithms (such as Augustus[39], Contrast[40], GeneID[41], NCBI gnomon, and SNAP[42]), and homology search engines (mainly BLAST-based), which perform well with long coding sequences, but less so with shorter ones[1,11,43], hence the lack of annotation of our set of translated lncORFs as coding sequences. More recent coding sequence predictors such as PhyloCSF[44] or AnABLAST[45], only identify as translated a very limited proportion of lncORFs (4.45% and 0.7%, respectively) detected as translated by Ribo-seq in this study (whether robust or limited) (Supplementary Fig. 1f), altogether highlighting the importance of experimental evidence to assess the translation of short ORFs.

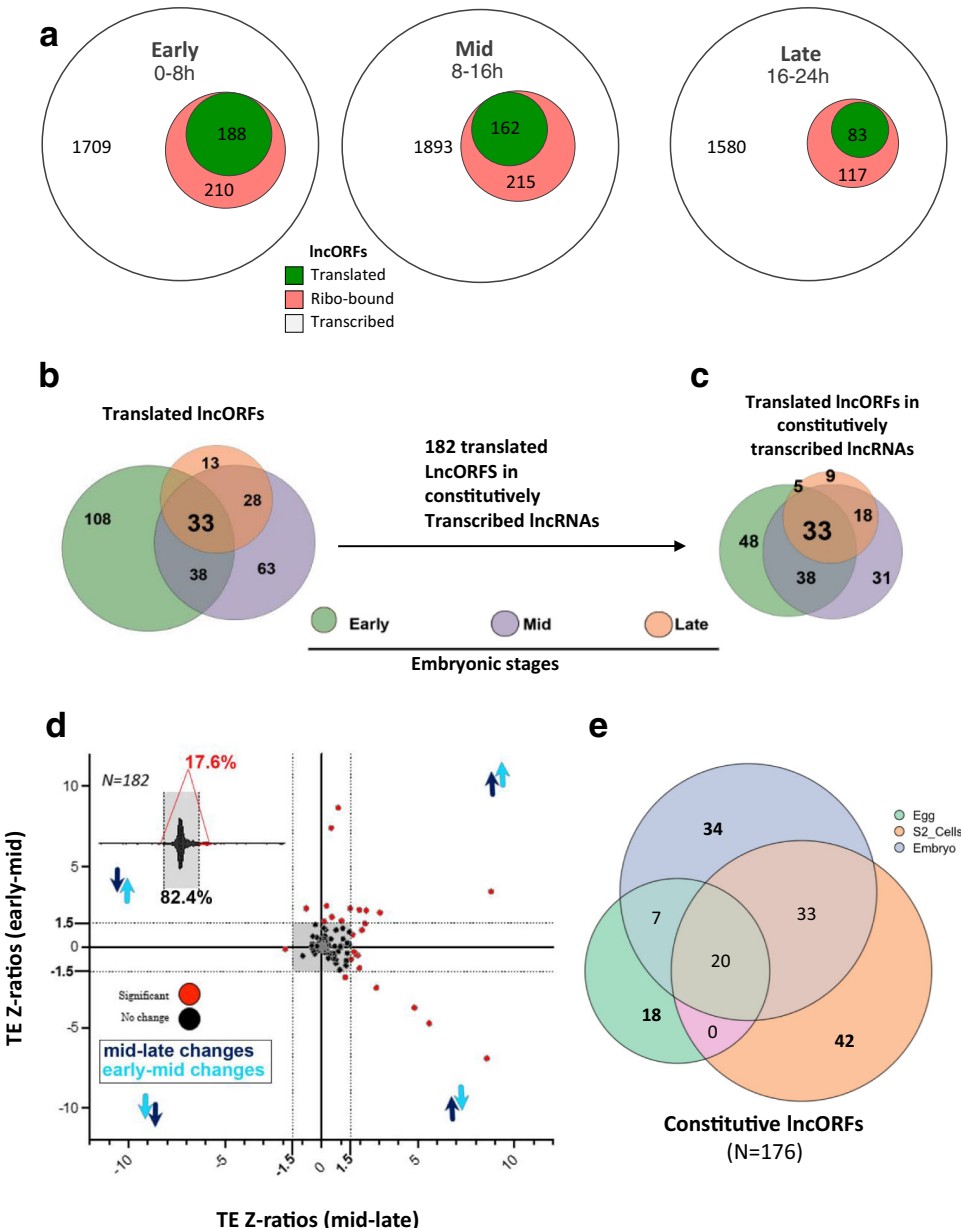

**Fig. 2 | Developmental regulation of lncORF translation. a** Number of transcribed (white), ribo-bound (red), and translated (green) lncORFs per embryonic developmental window. **b** Venn diagram showing lncORFs with translation signal per stage, from all lncRNAs. **c** Constitutively transcribed lncORFs are also translated in a stage-specific manner. **d** Quantitative fluctuations of lncORF translation across stages. TE (translational efficiency) Z-ratios across embryogenesis for constitutively transcribed lncORFs. Inset plot shows proportion of lncORFs with significant shifts in TE (17.6%) versus those with no significant TE changes. The majority (82.4%) of translated lncORFs with expression across embryogenesis show no significant quantitative modulation in translational efficiency across stages ($-1.5 \geq$ Z-ratio $\geq$ 1.5), for both analysed developmental window transitions (Early-to-Mid and Mid-to-Late). lncORFs with significant TE changes are highlighted in red. Arrows denote upregulation or downregulation from early-to-mid (light blue) or mid-to-late (dark blue) transitions. **e** lncORFs with constitutive transcription across different cellular contexts show high percentages of context-specific translation: Embryos (blue, top), 36% (34 lncORFs); S2 cells (orange, right) 44% (42); and Eggs (green, left) 40% (18).

## Developmental regulation of lncORF translation

We analysed how lncORF translation varies across the three different embryonic stages. lncORFs with translation signal in either replica appear to be more abundant in Early and Mid-stages, 188 and 162 lncORFs, respectively, compared to 83 in Late embryogenesis (Fig. 2a). Altogether, 259 lncORFs (89% of those translated at any stage) show stage-specific translation, while only 11% have constitutive translation signal (33 out of 292) (Fig. 2b). This pattern is similar to that of uORFs and differs from canonical ORFs, where 90% show translation, 67% of which is constitutive across stages[5]. This developmental regulation could involve transcriptional changes, changes in amount of ribosomal binding, or in the quality of the ribosomal engagement (framing). Transcriptional changes do not explain all stage-specificity, indicating that genuine translational regulation is involved. First, most translated lncORFs are transcribed in all stages (62%, 182 out of 292) (Fig. 2b). Second, translation of these constitutively-transcribed lncORFs also occurs in a stage-specific manner (Fig. 2c). Further, we observe that growing amounts of ribosomal association do not necessarily dictate productive translation, as indicated by framing (Supplementary Fig. 2d) suggesting that the amount of ribosomal association is not the only factor that favours lncORF translation, which supports the importance of the quality of ribosomal engagement.

Comparison with a sample of 182 canonical ORFs with similar characteristics (constitutive transcription and low ribosomal association levels (RPKM$^{FP}$)) corroborates that translational regulation is more prevalent amongst lncORFs (Supplementary Fig. 2a–c), and that the observed high levels of translational regulation are not simply due to random variations of low expression levels.

To further explore the basis of this translational regulation in development, we studied the contribution of qualitative changes in ribosomal binding (from translating (framed) to non-productive binding (ribo-bound only)). We observed that such qualitative changes occur in 38% (69 out of 182) of constitutively transcribed lncORFs. This is more than double the level observed in canonical ORFs, either amongst our lowly-expressed canonical control set (15%, 27 out of 182), or amongst canonical ORFs during the maternal to zygotic transition at early embryogenesis (14%[5]).

We next used the 182 constitutively transcribed lncORFs to quantify the role of changes in the intensity of ribosomal binding between successive embryonic stages with the translational efficiency metric (TE = RPKM$^{FP}$/RPKM$^{RNA}$, i.e. the ratio of translation to transcription). 17% (32 lncORFs) show significant changes in TE (Fig. 2d), while the rest show stable levels of ribosomal binding. Only 1% (2 out of 182) of lowly expressed canonical ORFs undergo this quantitative translational regulation (Supplementary Fig. 2a-c).

We extended our analysis to two different contexts: S2 cell cultures (derived from late *Drosophila melanogaster* embryos), and unfertilized *Drosophila melanogaster* eggs[31]. Although different, these contexts are closely related to embryos, yet significantly less complex, and thus provide other biological settings in which we can examine lncORF translational regulation. We identified 176 lncORFs with translation signal that are transcribed in all three contexts. A significant proportion of these lncORFs (44%, 36% and 40%, in S2 cells, embryo and egg lncORFs, respectively) showed context-specific translation signal (Fig. 2e). These results corroborate that qualitative translational regulation is a general feature of lncRNA translation.

## Translation is localized to a subset of polycistronic lncRNAs

The majority of transcribed lncORFs (2832 lncORFs in 602 lncRNA transcripts with RPKM$^{RNA}$ > 1) lack ribosomal association (RPKM$^{FP}$ > 1). Surprisingly, most lncORF ribosome-binding and translation events during embryogenesis are contained within 30% of all transcribed lncRNAs (602 lncORFs in 264 lncRNAs, Fig. 3a). The clustering of ribo-bound lncORFs in this 30% of lncRNAs is significantly different to what is expected at random (Fig. 3b). This suggests that a subset of transcribed lncRNAs is more prone to ribosomal binding than others. Strikingly, 77% of lncRNAs containing lncORFs with robust translation also contain other lncORFs with either ribosomal-binding and/or limited translation. Similarly, 63% of lncORFs with limited translation share their lncRNA with other lncORFs that display either ribosome-binding or robust translation (see below, Fig. 3e). Finally, there is a significant correlation between the number of ribo-bound-only (unproductive) and other lncORFs in *cis* displaying translation signal (limited or robust) (Pearson $r = 0.5998$, $p < 0.0001$) (Fig. 3c).

This clustering suggests that ribosomal association is a property of specific lncRNAs that in turn impinges on their lncORFs. That is, translation of a given lncORF seems contingent upon the ability of its lncRNA to engage ribosomes, and thus is linked to ribosomal-binding in other lncORFs in the same RNA. This concentration of ribosome-binding is not due to longer lncRNA length; in fact, lncRNAs without ribosome-binding can be significantly longer than those with robust translation (medians: 898 nt *vs*. 687, respectively; $p = 0.0198$) (Fig. 3d).

To corroborate that ribosomal association is preferentially displayed by certain lncRNAs regardless of context, and to exclude any effect of our PolyRiboSeq protocol, we generated RiboSeq data from lncRNAs from monosome fractions (RNAs bound by a single ribosome) of S2 cells, and analysed RiboSeq data from eggs[31]. The overlap of

translated lncORFs between monosomes and polysomes is very substantial (73%, Supplementary Fig. 3b), and their distribution of translation events per lncORF is also very similar (Supplementary Fig. 3e). These results rule out a protocol-specific bias, i.e. the observed clustering is not due to our method, PolyRiboSeq[5], focusing on polyribosomal RNAs ('Methods'). To further exclude a protocol or context bias, we studied Polysomal Profiling data, which consists of direct sequencing of RNAs located in Polysomal fractions[31]. We observe that lncRNAs containing translated ORFs in the embryo also show a very significant upregulation in polysomal RNA RPKM in S2 cells, with an average transcript RPKM of around 52.5, both in low (Fig. 3f) and high polysomal fractions (Supplementary Fig. 3c). Similarly, lncRNAs with embryo-translated ORFs also show significantly higher polysomal expression in eggs (Fig. 3f, Supplementary Fig. 3c).

Interestingly, some previously-characterized lncRNAs *bxd*, *Hsr-omega* and *Uhg1*, among others, are part of this group of lncRNAs with high affinity for polyribosomes, whereas others are not (*Uhg2*). Our translation analyses also show that transcripts from these lncRNA *loci* contain distinct lncORFs with different degrees of ribosomal association in *cis*: lncORFs with ribosome-binding without productive translation, lncORFs with limited translation, and robustly translated lncORFs. The *Hsr-omega* transcript also harbours miRNA sequences, and it has been previously shown that lncRNAs can act as miRNA precursors[46], and that miRNA precursor transcripts can encode functional micropeptides[47]. This prompted us to investigate the overlap between annotated miRNA sequences and our set of lncRNAs to see if the presence of miRNAs has any relation with the translation of lncRNAs. Overall, we found that 61 lncRNAs could act as miRNA precursors, of which only three also code for a translated lncORF (whether robustly or variable) (Supplementary Fig. 4a), suggesting that translated lncRNAs are in fact depleted of miRNAs, and thus, that these bifunctional transcripts are rare.

Altogether, data from three different techniques (PolyRiboSeq, RiboSeq and Polysomal profiling) indicate that lncRNA-ribosome association is repeatedly detected in a subset of lncRNAs, and that lncORF translation is associated with this property. This suggests a contingent model for lncORF translation, in which specific lncRNAs tend to associate with ribosomes, which is a necessary, but not sufficient, condition for eventual translation of some of their lncORFs in a polycistronic or quasi-polycistronic manner, with the identity of particular lncORFs to be translated due to an interplay between intrinsic and extrinsic factors (see section 'Developmental regulation of lncORF translation' above and 'Molecular factors driving lncORF translation' below).

## Molecular factors driving lncORF translation

To better understand the mechanisms underlying individual lncORF translation, we analysed different features and correlated them to the mode of translation for individual lncORFs. A generally good predictor of translation is a higher amount of ribosomal binding relative to transcription, with translated lncORFs showing a similar distribution of translational efficiency (TE) values to that of canonical coding genes and translated uORFs[5] (Fig. 4a).

We have also analysed the 7-nucleotide Kozak sequences, which are known to influence translational initiation around the start codon[48,49] (Fig. 4b, Supplementary Fig. 4b). Robustly translated lncORFs show Kozak sequences significantly closer to the optimal consensus ($p = 0.0364^*$ *Vs* limitedly-translated lncORFs; $p = 0.0079^{**}$ *Vs* Ribosome-bound lncORFs). Limitedly translated lncORFs show a Kozak score similar to those which are only ribosome-bound ($p = 0.6798^{NS}$), indicating that a component of their variability could be weak Kozak sequences. Interestingly, there is no correlation between RPKM$^{FP}$ and Kozak context (Pearson $r = 0.03716$, $p = 0.3627$), whereas Kozak context has a small but quite significant correlation with framing (Pearson $r = 0.1054$ $p = 0.0096^{**}$). Hence, the Kozak

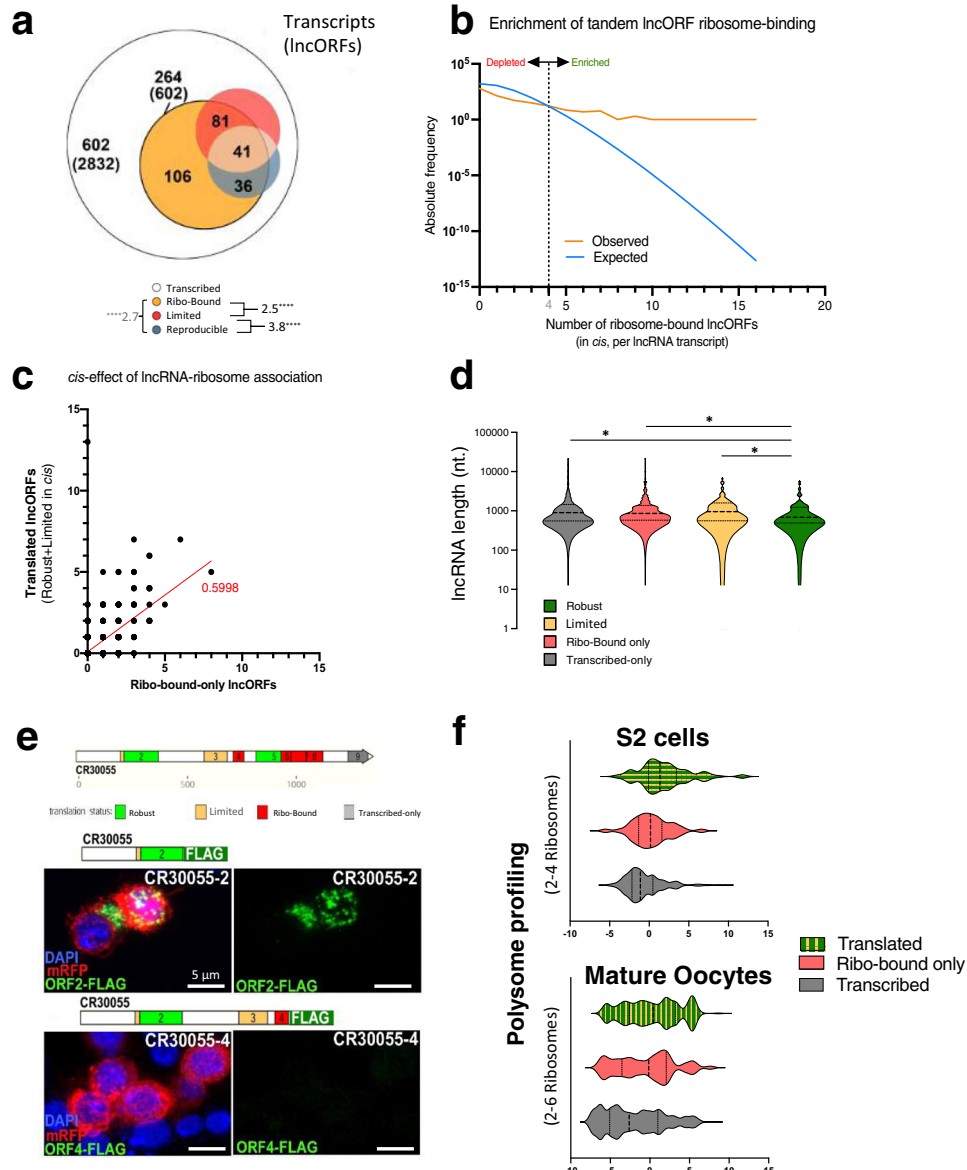

**Fig. 3 | Clustering of translation in lncRNAs. a** A subset of lncRNA transcripts shows accumulation of translation events in *cis*. Top, proportional Venn diagram representing lncRNAs according to translation signal detected within their lncORFs (parentheses= lncORF numbers). Bottom, pairwise overlap comparisons, showing corresponding representation factors, and their significance (****$p < 0.0001$, representation factor analysis, see 'Methods'). **b** Number of lncORFs in the same lncRNA showing ribosomal binding (RPKM$^{FP} > 1$, orange), compared with expectations as given by a Poisson model (blue). Values to the right of their intersection (dotted line) show the enrichment of *cis*-related binding. **c** Correlation between ribosome-bound-only and translated lncORFs in the same lncRNA. Pearson's $r = 0.5998$. **d** lncRNA length does not explain clustering of ribosomal binding and translation events to particular lncRNAs. Violin plots of annotated lncRNA transcript lengths (nt) in function of the translation signal detected within their lncORFs. $N = 866$ transcribed lncRNAs (see panel 3a). "*" denote *p*-values <0.05. $p = 0.0133$ for "reproducible-variable" comparison; $p = 0.0183$ for "reproducible-ribo-only" comparison; $p = 0.0198$ for "reproducible-transcribed" comparison. Mann–Whitney tests, two-tailed. **e** CR30055 is an example of a lncRNA with multiple ORFs: ORF2 appears as robustly translated by Ribo-seq, and ORF2-FLAG shows expression in S2 cells, whereas ORF4, appears as ribo-bound-only, and shows no expression in S2 cells, despite sharing the same transcript as ORF2. **f** Polysomal RNA RPKM values of lncRNAs from low polysomes in S2 cells (2–4 ribosomes per lncRNA, top) and Eggs (2–6, bottom) are enhanced for embryo-translated lncRNAs, suggesting that translated lncRNAs have an intrinsic higher affinity for ribosomes.

sequences may mostly influence the recognition of specific AUGs to guide in-frame lncORF translation, consistent with its known role in promoting ribosomal assembly during canonical initiation, whereas the net amount of ribosomal association (RPKM$^{FP}$) must involve additional factors, such as 5′ cap and 3′UTR sequences.

We find a striking correlation between the translation status of a lncORF and its 5′ to 3′ position in *cis* within its lncRNA transcript, in relation to other ORFs (Fig. 4c). While lncORFs which are transcribed-only show a median position of 7th in a transcript, ribo-bound-only ones show a median position of 4th. Limitedly translated ORFs show a

lower median position, 3rd, while for robustly translated the median position decreases to 2nd. In the latter case, more than a third of lncORFs are the first ORF within their transcript. This striking position effect is also observed in monosomes (Fig. 4d), suggesting that it influences the start of lncRNA translation. This position effect on translation potential fits a model of 5′ to 3′ re-initiation of translation for polycistronic transcripts, as observed in yeast and suggested for *tarsal-less*[8]. With this mechanism, translation can re-initiate 3′ (only) of a translated ORF, but with diminished efficiency as not all ribosomes re-initiate translation at the 3′ORF[50,51]. We corroborated this

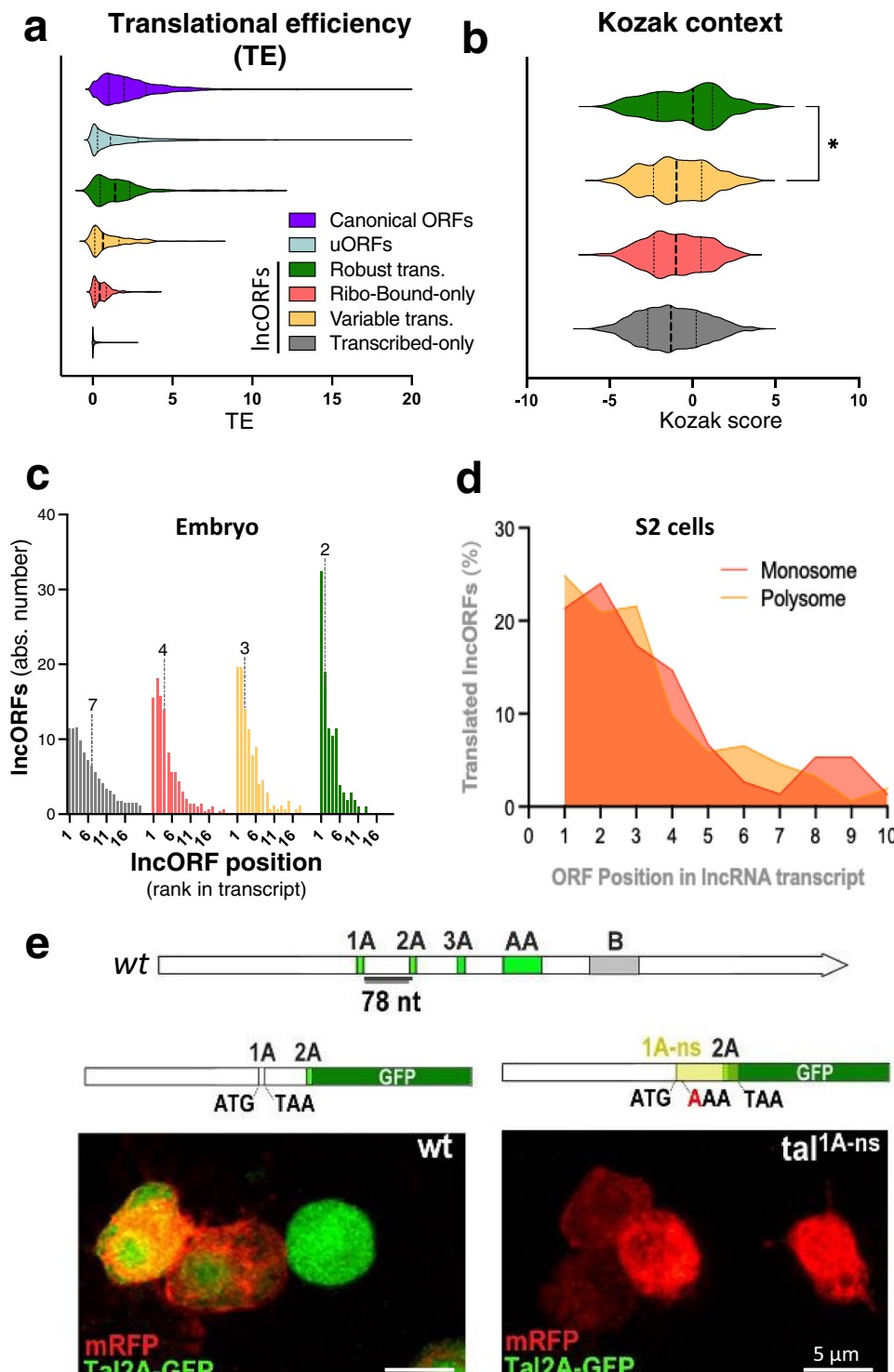

**Fig. 4 | Expression levels and *cis*-factors affect lncRNA translation.**
**a** Translational efficiency across lncORF categories. Violin plots for Translational efficiency values (TE, average across embryogenesis) for all lncORFs in each translation class, and for canonical annotated ORFs and uORFs (data from Patraquim et al.[5]). Thick dotted lines denote median, thin dotted lines denote lower and upper quartiles (note the more modest changes in RNA levels - Supplementary Fig. 3a). **b** Kozak sequences, scored against canonical ORF consensus sequence, for lncORFs with different translation status (0 = canonical average; colours as in **a**). Robustly translated lncORFs have Kozak sequences significantly closer to canonical ORFs. $N_{Transcribed-only} = 2701$; $N_{Ribo-bound-only} = 310$; $N_{VariableTransl.} = 185$; $N_{Robust} = 107$.

"*" denotes $p < 0.05$ (exact $p = 0.0364$. Mann–Whitney tests, two-tailed).
**c** Frequency distribution of relative lncORF positions in their lncRNA (cistronicity) (1 = closest to 5′ end), per translation class (colours as in **a**). Dashed lines and numbers denote median values. **d** Similar cistronic position of translated lncORFs in monosomes (dark orange) and polysomes (light orange). **e** Translation of ORF 2A within the polycistronic *tal* transcript is lost upon removal of the Stop codon of the upstream ORF *1A* (*tal[1A-ns]*), which extends ORF 1A beyond the stop codon of 2A (yellow), leaving no chance for 5′-to-3′ re-initiation to occur. ATG: START codon. TAA: Actual STOP codon. AAA: mutated STOP codon. Source data are provided as a Source data file.

mechanism in the well-characterized polycistronic *tarsalless* transcript[5,8,52,53], where we extended the tal1A ORF over the downstream tal2A ORF, by removing the tal1A stop codon, so that the resulting ORF only stops after the tal2A start codon. This modification completely precludes tal2A translation, since, in this case, ribosomes only re-initiate downstream of the tal2A start codon (Fig. 4e, Supplementary Fig. 4c).

More surprising was the lack of effect of codon and aminoacid usage on translation signal, since usage of rare tRNAs and AAs can delay translation[54]. Although lncORFs in general display non-canonical AA usage (Supplementary Fig. 4d, see also refs. 3, 11), and sub-optimal codon usage (Supplementary Fig. 4e), we observed no significant differences between the translated and the ribo-bound populations in either metric (Supplementary Fig. 4d, e), indicating that translated lncORFs are not evolutionarily fine-tuned for coding efficiency, possibly reflecting their low resource requirements, given their generally modest translation levels and developmentally restricted expression patterns, and/or recent evolutionary origin.

### Phylogenetic conservation supports lncRNA translation and micropeptide function

ORFs from non-coding regions have been posited to have arisen recently and be neutrally evolving[55]. However, assessment of conservation depends on the ability to identify orthologues. Our group has previously shown that standard homology-detection methods such as BLAST are not adequate for smORFs[1]. Based on successful experiences at detecting and experimentally corroborating smORF orthologues, including putative lncRNAs[8–10,20,56,57], we have developed GENOR 1.0, a bioinformatic pipeline for smORF homology detection (Fig. 5a and 'Methods'). Briefly, it relies on recurrent *jackhammer* searches to identify potential smORF homologues, using reciprocal searches and MAFFT alignment scores for hit validation. We have applied this pipeline to in silico translations of extensive transcriptomic data available for 12 Drosophilid species.

Firstly, we have validated this pipeline by analysing all 862 annotated shortCDSs, i.e. smORFs already annotated as coding, which tend to encode peptides of around 80AA in length in monocistronic mRNAs[11]. GENOR detects homologies for 666 of the 670 sCDS with annotated orthologues, often extending the homologies to more distant species. Further, of the 191 smORFs with no annotated homologies in Flybase, GENOR-detected homologues for 186 (identity threshold >50%, Fig. 5b, c, Supplementary Fig. 5a). Of these, 153 show homologues in more than one species (Fig. 5b. These results suggest that GENOR reaches to and beyond standard methods in detecting homologies for small open-reading frames.

Applying our pipeline to lncORFs, we observe that while they are lowly conserved in general, those with ribosomal association can be conserved within the *Drosophila* genus (Fig. 5d, f, Supplementary Fig. 5b, see also Supplementary Dataset 1). Within this group, robustly translated lncORFs show a significantly higher degree of conservation than those purely ribo-bound. In other words, depth of conservation (ORF age) correlates strongly with translation status (Fig. 5d). This conservation can extend across the genus, but is more often limited to sister-species of *Drosophila melanogaster* (*D. simulans* and *D. sechelia*), and is compatible with translated lncORFs appearing recently as novel coding sequences (genes) within the *Drosophila* genus (Supplementary Fig. 5b). In these species, the overlap between lncORF groups which are either conserved (as per GENOR) or robustly translated (as per framing analysis) is statistically significant and noticeably high for robustly translated lncORFs (Supplementary Fig. 5c), indicating that the previously-observed correlation between translation status and conservation is not expected in a random sample—with an over-representation between 4 and 8 times that which is expected by chance. This discards passive conservation due to phylogenetic closeness as an explanation for these results. A few examples of these

robustly translated and conserved lncORFs (*Drosophila simulans)* can be consulted in Supplementary Fig. 5d.

However, conservation of coding ORFs does not imply a function for the encoded peptide. For example, uORFs can have a non-coding function as *cis*-translational regulators, mediated by ribosomal binding and capture, and independent of the encoded peptides. To ascertain conservation of coding function, we studied the pattern of nucleotide substitutions within lncORF sequences, and in particular whether there is an evolutionary signature of natural selection acting on these coding sequences. Our dN/dS analysis shows significant prevalence of synonymous substitutions that preserve the coding sequence in translated lncORFs, while this effect does not appear significantly in ribo-bound or transcribed only lncORFs (Fig. 5d, see also Supplementary Dataset 2). This effect is not an artefact due to a biased and more intense substitution rate in translated lncORFs, since the net amount of nucleotide conservation is similar across all lncORFs (Supplementary Fig. 4f). Again, this observation suggests that translated lncORFs are being selected.

A final observation in line with the evolution and selection of translated lncORFs is ORF length. A comparison of ORF lengths, shows that Robustly translated lncORFs tend to be significantly longer than ribo-bound only ORFs (Supplementary Fig. 4g), an interesting observation since larger peptides could be potentially more stable than shorter ones and thus more likely to convey a biological function. This observation is not an artefact but likely reflects the evolutionary selection of translated sequences; first, our binomial pipeline is not length-dependent[5]. Second, lncORFs have expected random lengths[11], whereas Robustly translated lncORFs are twice as long as ribo-bound ones, which is highly improbable to occur simply by chance.

## Discussion
### lncRNA translation
Applying Ribosomal profiling with extensive sequencing depth and increased resolution of framing revealed the dynamics of lncRNA association with ribosomes. lncRNA translation can be reproducibly observed in 30% of lncRNAs, in at least two independent biological samples, each including several independent translation events. Further, this translation can be reproduced ex vivo using tagged ORFs in cell cultures. Finally, the molecular signatures of this translation (size of RNA footprints protected by ribosomes, favoured codon framing) are identical to ORFs in canonical mRNAs, such that it is generally not possible to distinguish whether an individual ORF data identifies a canonical ORF, or a lncORF. The exception might be intensity of ribosomal binding and translation, which is generally lower for lncORFs, despite cases with strong canonical-like signal. We conclude that lncRNA translation is a replica of the canonical one, albeit at lower intensity and with a lower chance of being constitutive, factors which have precluded its unambiguous identification and characterization to date. We note that other studies have relied on smaller sample sizes (both in terms of absolute number of RiboSeq reads, number of reads mapped to the genome, and reads used to determine framing)[31,36]. Such "average" sampling, although adequate for canonical mRNAs, may not allow reliable detection of lncRNA translation; we recommend that a minimal depth/coverage of 40 Million mapped reads mapped per -1000 transcribed lncRNAs must be used per replica.

lncRNA translation appears to be a two-step process influenced by specific factors (Fig. 6a): Firstly, some lncRNAs have a tendency to locate to polysomes and associate with ribosomes, as revealed by PolyRiboSeq and Polysomal profiling. The clustering of lncORFs in the same lncRNA showing either robust or limited translation, or ribosomal-binding-only, can be explained by the affinity of their lncRNA for ribosomes. In a context of circularized RNA translation, with each RNA undergoing several cycles of translation by the same ribosomes, ribosome capture by the RNA 5'cap, and subsequent ribosome engagement and retention by some ORFs can benefit all other ORFs in

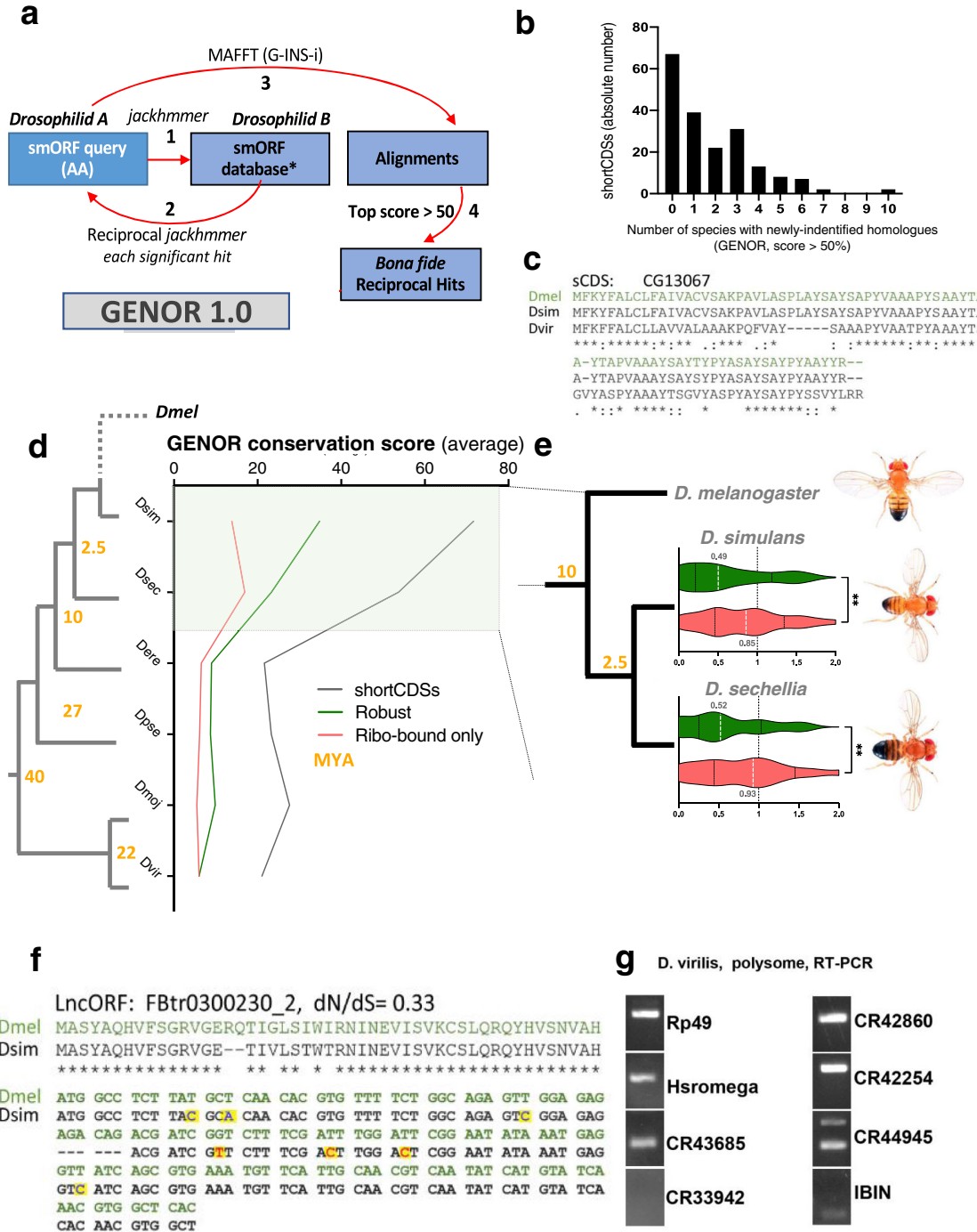

**Fig. 5 | lncORF sequence evolution across *Drosophilids* and emergence of novel coding genes. a** GENOR pipeline for the detection of smORF evolutionary conservation across the *Drosophila* sp. genus. Each smORF is used to query, via jackhammer, smORF databases obtained from available RNA expression data for each target species, ensuring that putative homologues come from transcribed genes. Matching reciprocal hits are aligned using MAFFT, with a smORF-calibrated threshold score deciding on the conservation status of the top hit per ORF (see 'Methods'). **b** Number of sCDSs with homologues identified by GENOR, plotted against the number of *Drosophila* species in which those homologues were identified. **c** AA sequence alignment of CG1307, a sCDS lacking annotated homologues, and its GENOR-identified *Dsim* and *Dvir* homologues. **d** Average conservation score across robustly translated or ribo-bound-only lncORFs, and translated sCDSs, for 6 species across the *Drosophila* phylogeny. Pale green rectangle: phylogenetic distance with substantial conservation signal for translated lncORFs as detected by GENOR. Yellow: Million Years Ago (MYA). **e** Robustly-translated lncORFs show evidence of purifying selection. Distributions of dN/dS values measuring natural selection (fraction of non-conservative nucleotide changes vs. nucleotide changes conserving the AA sequence) acting on robustly translated (green) or ribo-bound-only (red) lncORFs, in pairwise ORF alignments between *Dmel* lncORFs and syntenic ORFs in either *Dsim* or *Dsec*. "**" denote *p* < 0.01 (exact *p* for Dmel-Dsim=0.0074; exact *p* for Dmel-Dsec = 0.0029). Mann–Whitney tests, two-tailed. **f** AA and nucleotide sequence alignments of lncORF FBtr300230_2, within the *uhg4* transcript, and its *Dsim* homologue identified by GENOR, showing a great extent of AA conservation, and a pattern of nucleotide changes (dN/dS score: 0.33) consistent with a coding function for this lncORF. Substitutions are highlighted by yellow squares. Blue: Synonymous substitutions, Red: non-synonymous ones. **g** GENOR-detected homologues of translated *Dmel* lncORFs in *Dvir* or beyond, are loaded into polysomes in *Dvir*, suggesting that their translation might also be conserved (full gel with molecular weights available in the "Source data" file of this manuscript). *Dmel*: *D. melanogaster*. *Dsim*: *D. simulans*. *Dsec*: *D. sechelia*. *Dere*: *D. erecta*. *Dpse*: *D. pseudoobscura*. *Dmoj*: *D. mojavensis*. *Dvir*: *D. virilis*. Source data are provided as a Source data file.

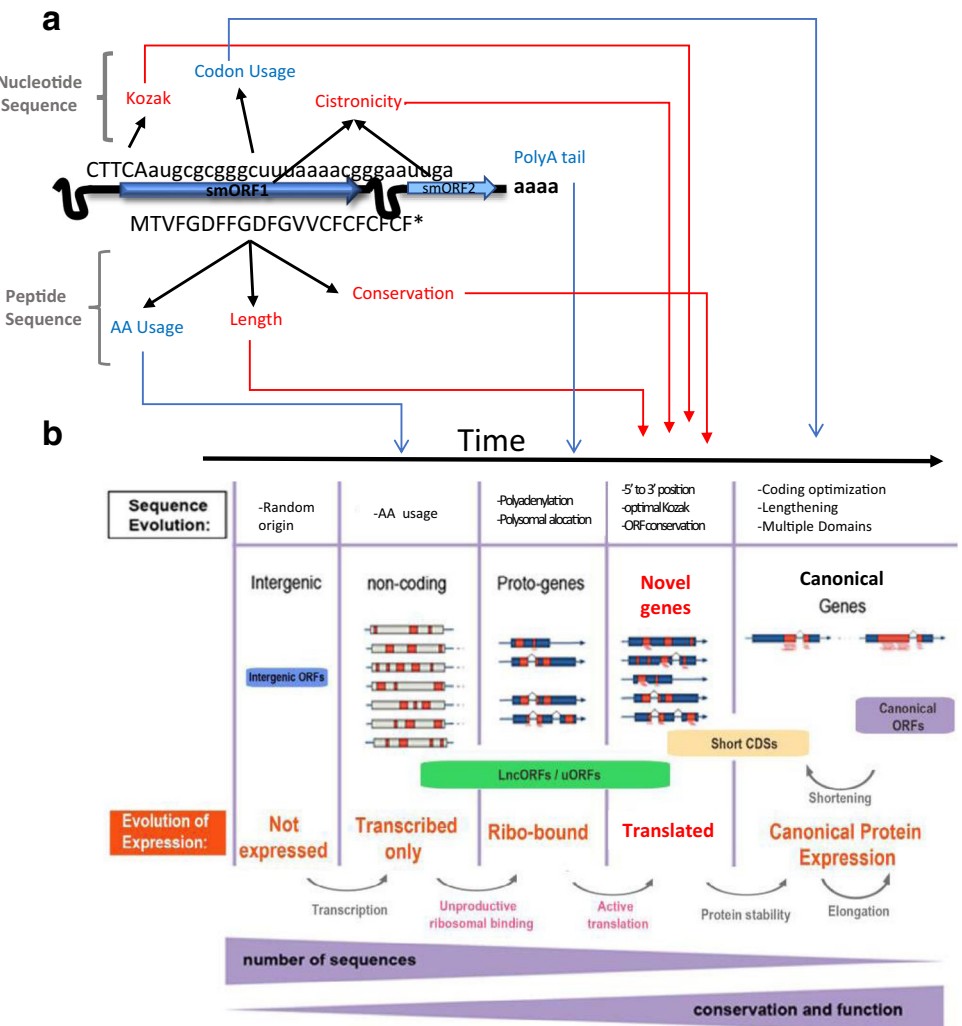

**Fig. 6 | Model for the activation of lncRNA translation, and the evolution of novel coding genes. a** Molecular features directly linked to lncRNA translation (red) and those linked to related processes (blue). **b** Evolutionary acquisition of lncORF translation by accretion of molecular features from **a**), leading to the emergence of novel coding genes. Data from this and other works[5,11].

the transcript. The particular amount of translation would be dictated by Kozak goodness-of-fit, and by 5′ position, which suggests a re-initiation mechanism as displayed by *tal*, with diminishing 5′ to 3′ efficiency.

Ribosomal binding must require specific RNA properties such as stability, cytoplasmic localisation, poly-A tail, and an adequate RNA length, and its primary function does not need to be translation, but some regulatory activity, either on other transcripts, or on lncRNAs themselves. It could even be possible for polysomal localization of some lncRNAs to initially be a mostly irrelevant secondary consequence of a non-coding function: for example, polyA tails may enhance lncRNA stability, and polysomal localization might be an unintended consequence of little functional significance for the cell. However, there is evidence for polysomes being an active site for lncRNA degradation[58], which would provide a non-coding functional relevance. A similar source of ribosomal binding could be a pioneering, non-productive round of ribosomal scanning as part of NMD proofing. Finally, ribosomal binding could have a regulatory or structural role on non-coding functions of the lncRNAs. uORFs have been shown to *cis*-regulate the translation of canonical ORFs in the same mRNA, either negatively[59] or by stabilizing it[5].

In a second step, within these ribosome-bound lncRNAs, the translation of particular lncORFs is favoured by specific molecular features, such as intensity of ribosomal engagement and retention in the ORF (revealed by TE), optimal Kozak sequences to guide AUG initiation, and finally, access to ribosomes, dictated by 5′ position within the transcript, since efficiency of 3′ re-initiation is lower. Although a role for non-AUG translation has been suggested in shaping translation programs, we have not addressed it in the present study, as its low efficiency, when compared to AUG START codons, is expected to significantly hamper its detection in the case of lncRNAs. It, however, conceivable that some of these lncORFs which are Ribo-bound or show limited translation are in fact robustly translated but lack framing due to leaky ribosomal occupation signal coming from an overlapping non-AUG lncORFs[60]. Finally, developmental stage-specific factors must take advantage of these features for translation to take place in a regulated manner. These factors could be, for example, relative availability of specific translation factors, or specific ribosomal components or tRNAs (since, according to codon usage, lncORF tRNA requirements are different to those of canonicals). Expression of either robust or limited translation may also depend on a combination of intrinsic (RNA and ORF) and/or context-dependent features. Thus lncORFs having limited translation at embryogenesis could display robust translation at other life stages or under non-standard environmental conditions.

These observations again fit a canonical mode of translation, with the addition of polycistronic re-initiation (Kozak). Robust translation

of a 5′ lncORF can result in ribosomal binding and/or limited translation of another lncORF located 3′ by virtue of re-initiation, occasionally producing polycistronic translation. Here we prove the mechanism (re-initiation) for the polycistronic translation of *tal*, an ex-lncRNA that we showed by genetics to encode functional peptides[8], and by RiboSeq to be translated[5].

## lncORF evolution

The molecular characteristics favouring lncORF translation could be initially random (Fig. 6b), and then preserved or improved in lncORFs encoding advantageous peptides, as indicated by the conservation of their AA sequence across evolution. In turn, the observed instances of purifying selection are strong proof that those encoded micropeptides are translated and functional. If they had no function, their AA sequences would not have been selected and conserved; but to be selected, these sequences needed to be exposed to natural selection, i.e., they must have been translated into peptides.

Robustly translated lncORFs are either conserved across short evolutionary distances, or seem to arise de novo in *D. melanogaster*. Hence, they look as expected of novel coding genes, fitting with the fact that their codon usage has still not evolved into canonical optima, and with their overall levels of nucleotide evolution (revealed by Phylo-P (this work), PhastCons[3], and CIPHER[30]. In turn, limitedly translated, and ribo-bound-only lncORFs behave as protogenes, i.e. sequences with sub-threshold ribosomal association and stochastic translation, forming a reservoir of pre-adaptations awaiting favourable conditions to become de novo genes. These conditions could be sustained stress or other changes in the cellular or organismal environment, leading to changes in gene expression and translation in particular organs or moments of the life cycle. The observed susceptibility of lncORFs to translational regulation could favour this process and, in turn, the acquisition of functional niches for the new peptides.

For most of our bioinformatic markers, we observe a continuum ranging from transcribed-only, to ribo-bound-only, limitedly translated, and finally robustly translated lncORFs; and then, from robustly translated lncORFs to short CDSs and canonical ORFs[5,11]. This includes (a) molecular markers of translation; (b) AA conservation, and (c) encoded peptide characteristics, such as length and AA usage (see also[11]). These correlations suggest that this continuum is evolving in time, from non-coding sequences to canonical coding genes, and thus giving rise to novel coding genes (Fig. 6b). The transition from non-coding lncORFs to coding ones seems to take place within a genus, i.e. in a time-scale of millions to tens of millions of years. This time-scale fits with observations in unicellular eukaryotes[30], but must be further tested, and the molecular processes involved further defined. For example, the transition non-coding/coding seems to take place within potentially polycistronic RNAs, be it lncRNAs (this paper) or via uORFs in mRNAs[5,14]. Borrowing a concept we proposed for uORFs, it seems that lncRNAs also behave as "gene nurseries". However, unicellular eukaryote genomes include uORFs, but to a lesser extent lncRNAs, and so the mechanisms for "gene birth" may have diversified during evolution.

## Function of lncRNA translation

Having established the principle of lncRNA translation, first by experimentally proving the coding function of specific lncORFs[8,10,12,56,61,62], and then by extending this finding to the genomic level[6,63,64] and this study), the next question is the actual molecular function of the translated ORFs, and which perform their cellular role at the level of encoded and translated micropeptides. We find that robustly translated lncORFs as a group contain clear evidence for the conservation of AA sequences, an indication that a good proportion of this cohort is exerting its function at the protein level. Micropeptide function does not preclude a co-existing, or even synergistic non-coding function for the encoding lncRNA. For example, uORFs have been shown to stabilize the translation of their *cis*-downstream

canonical ORFs[5], but also to produce micropeptides that bind and functionally cooperate with the protein encoded by such *cis*-canonical ORFs[14]. Similarly, micropeptides produced by a lncRNA have been shown to enhance the function of miRNAs produced by the same lncRNA[65], but we observe that in *Drosophila melanogaster*, this type of dual function must be rare. Alternatively, it would be possible for lncORF micropeptides to have a function independent of their encoding lncRNA; and finally, that the predominant or only function of some lncRNAs is to produce functional micropeptides. Indeed, there has been repeated evidence of genes classified as non-coding, only to be revealed as encoding functional micropeptides with vital and diverse functions in multicellular eukaryotes, be it plants, vertebrates or invertebrates[19–22]. A general theme in micropeptide function is their ability to bind and regulate canonical proteins, a function well suited to their small size, which also probably limits structural roles. Here we observe translation in lncRNAs with characterized non-coding function, but only further studies can distinguish amongst these possibilities.

However, the most important function of lncRNAs might not be actual, but potential: their ability to act as "gene nurseries" to provide our genomes with new genes to evolve, and thus not be limited to recycling existing coding sequences when trying to produce new adaptations to both current and new challenges. In turn, we see enticing possibilities in the study of potential abilities for lncORF "proto-genes": for example, lncORF micropeptides have "exotic" AA sequences not observed in canonical proteins[11] (Supplementary Fig. 4c) and while antimicrobial activity is a common function of sCDS peptides[10,11,66], untranslated lncORF peptides with antimicrobial potential should encounter no natural resistances if they were artificially expressed.

To clarify these possibilities for hundreds of lncRNAs and lncORFs in each species will likely take a determined experimental effort for the foreseeable future. Here we could not assess the translation of the 1679 lncRNAs not showing transcription in embryos, and further, given the variability in lncRNA translation, it is also possible that some of the lncRNAs that we observe as not translated during embryogenesis may be translated in another context in *Drosophila melanogaster*, or in a related species. Given that many characteristics of lncRNAs we have tested are also found in vertebrates and plants[11], we expect this to apply to eukaryotic metazoans. By extrapolation from our data, which reveals up to 0.3 micropeptides per transcribed lncRNA, we estimate that around 7000 non-annotated, novel micropeptides exist in humans. Thus, while these uncertainties in lncRNA function continue, we believe that to maintain a blanket classification of thousands of RNAs in each genome as "non-coding" in the absence of relevant data, can only hinder our scientific understanding of these sequences. In search of scientific precision, and to avoid confusions, a solution might be to preserve the lncRNA acronym, yet change its meaning to 'long non-canonical RNAs', which does not prejudice their function.

## Methods

### RNA sequencing and quantification of gene expression

**RNA-Seq and Poly-Ribo-Seq.** We first used our published embryo RNA- and Poly-Ribo-Seq samples[5], gathering all previously-obtained reads and performing additional sequencing from the same samples—822 million additional raw Poly-Ribo-Seq and RNA-Seq reads, obtaining a further 77 million genome-aligned ribosome footprints and 39 million genome-aligned cytoplasmic RNA reads, for a total of 570 M Riboseq and 256 M RNAseq reads. All novel sequencing of embryonic stages was thus conducted on previously-collected biological samples and pooled with all previously-published data of the same biological sample and protocol, after bioinformatic processing with the pipeline described previously[5]; genome-alignment and quantification of ORF expression in both experimental conditions were recalculated for the complete sequenced set, using the previously described method.

**Polysomal profiling.** Genome-alignment and gene expression analyses of previously-published polysomal profiling datasets were performed in the same way as for others sequenced for this publication. The datasets analysed were published elsewhere, and are available online[3,31].

**lncORF prediction and selection**
All ncRNAs from Flybase release 6.13 (*dmel-all-ncRNA-r6.13.fasta*) were scanned for putative ORFs (AUG-STOP) with a minimum of 10 codons, using EMBOSS *getORF* (*getorf :find 1 -reverse No -minsize 30 -maxsize 450*), yielding 22,262 ORFs. The genomic coordinates of this set of long non-canonical RNA ORFs (lncORFs) was then compared to eliminate duplicates and internal ORFs, resulting in a set of 18,507 ORFs with unique coordinates. Finally, the coordinates of these ORFs were compared with other ORF classes (command *bedtools intersect -v -s -a*; no overlaps with both annotated coding sequences and uORFs) to obtain a set of 16,335 unique, non-overlapping lncORFs for further analysis. ORF sets from other RNA classes were previously defined[5].

**Comparison of framing across ORF classes**
Ribosomal rRNA-depleted reads (bioinformatically) were aligned to the complete predicted ORFome of *Drosophila melanogaster*. For each ORF in the set, the AUG-STOP region was included, as well as the −18 and +15 nucleotide stretches surrounding the ORF. ORF classes were then separated and analysed individually for global framing patterns, using the riboSeqR package; all RFPs of 26−36 nucleotides in length were used for this, as well as subsequent framing analyses. Per RPF length, the dominant frame across each ORF class was defined as the properly translated frame, with the remaining two as noise. This allowed us to compare global framing patterns across classes using Spearman's rank-order correlation.

**High-resolution framing**
Framing patterns on each ORF were evaluated separately for each RPF length of 26−36 nucleotides using a binomial test with $p < 0.01$[5]. The frame considered translated in each RPF length was defined as the dominant frame in the ORF class (see above). ORFs with read patterns that passed the binomial test in a given RPF were considered phased or framed.

**Defining a set of bona fide translated ORFs**
Per stage, we defined a translated ORF as having RNA-Seq RPKM > 1 in one of two replicates (transcribed), as well as Poly-Ribo-Seq RPKM > 1 in both biological replicates (Ribo-bound) and framing in at least one RPF length (framed). If the last condition was not met, the ORF was considered Ribo-Bound-only, with no productive translation observed. In the event of transcription being the only signal in a given ORF, we defined it as "Transcribed-only". This allowed for the definition of sets of Transcribed, Ribo-bound as well as Translated ORFs per stage. Within the Translated set, ORFs showing framing in one replica only were considered 'limited' and those framed in both replicas as 'robust'.

**Cloning**
Our strategy for ORF tagging respects RNA and ORF contexts, introducing 3′ tags to lncORFs while maintaining their 5′UTRs. The 5′-UTR and CDS of selected lncORFs, were obtained by PCR, from embryonic cDNA and cloned into pAWF whose FLAG ATG start codon was mutated to GCG. Cloning was performed by NEB Hi-Fi assembly, digesting the pAWF vector with EcoRV and AscI, and using primers with overlapping regions, which reconstituted the FLAG sequence.

**S2 cell culture, transfections and imaging**
S2 cells were grown under standard conditions in Schneider´s medium with 10% FBS. Transfections, were carried out with Effectene (QIAGEN), on poly-l-lysine treated coverslips, in 12-well culture plates, seeding 500,000 cells in 800 μL of media, and using 300 ng of DNA per transfection (100 ng Act5-lncORF-FLAG, 100 ng act5-gal4, 100 ng UAS-mCD-RFP8). After 48 h, cells were fixed for 20 min with 4% formaldehyde, washed with 1X PBS, 0.1% Triton X−100 (PBTx), blocked with PBTx, 2% wt/vol BSA before immunostaining with primary mouse anti-FLAG M2 antibody (Sigma) at 1/1000 and secondary anti-mouse FITC (Jackson, West Grove, PA) at 1/500. All transfections were incubated for 10 min with DAPI (Sigma) according to manufacturer's instructions for nuclei staining and mounted with Vectashield (Vector Labs, Burlingame, CA). Imaging was conducted using a Leica 63X Plan Apochromat Oil Immersion lens on the Leica Stellaris confocal microscope, acquiring 4-tile arrays of 350 μm/350 μm, and Z-stack images taken with a 1.6 μm slice interval (6 slices/array). Image J and standard plugins (v 1.53c) were used for quantifications of FLAG signal. For each array, a thresholded image after merging all colours was used to create a mask in order to map all cells within the array as particles (using particle analyser plugin, sizes 100 to infinite). For each particle, the average green and red intensities were measured. For each experiment we plotted the top 30 values for green signal (FLAG), normalised by average top 30 red signal (mCD8-RFP) values (to take into account possible variation in transfection efficiency between experiments). R (v.4.0.3) and GraphPad Prism (v.9.1.1) were used for statistical analyses.

**Quantitative changes in translation efficiency**
We used Z-ratios to assess significant quantitative changes in translational efficiency per ORF. Briefly, Translation efficiency (TE) per ORF was first calculated as the fraction of normalised ribosome binding reads ($RPKM^{FP}$) over the normalised RNAseq reads ($RPKM^{RNA}$). Then, we performed Z-ratio calculations of TE variation across time-contiguous developmental stages (from Early to Mid, and from Mid to Late), using a previously-suggested empirical score of ±1.5 to define significant regulatory events[5,67].

**Translation markers in *cis***
**Kozak context scoring.** For all canonical ORFs, we extracted the nucleotide composition around but excluding annotated START codons (positions −5 to 6, but excluding nucleotides 1−3). For each position, we then calculated the *log* odds ratio between observed and background nucleotide frequencies−the latter calculated as position-independent relative frequencies of the same sequences). This provided a scoring table of position-specific nucleotide frequencies from bona fide *Kozak* contexts with which to score individual lncORFs. The final Kozak score per ORF was then obtained by adding the individual position-specific values for all observed nucleotides.

**ORF position in *cis*.** lncRNA ORF position in *cis* was defined as its absolute position rank) in the 5′−3′ strand per annotated transcript among all predicted ORFs (see above in 'lncORF prediction and selection'), regardless of transcription translation status.

**Codon usage.** Codon usage bias was calculated per lncORF using the MILC metric (R package *coRdon*[68]), using similarly-calculated codon biases for all annotated canonical ORFs as control set.

**dN/dS**
To evaluate any signatures of natural selection acting on our lncORF set, we applied a dN/dS test to pairwise alignments using the *dnds* function in the R package *ape*, which is an implementation of the metric published in Li et al.[69]. The R package *seqinr*[70] was used to read syntenic pairwise alignments per ORF (*Dmel-Dsim*, *Dmel-Dsec*, *Dmel-Dyak* and *Dmel-Dere*). All syntenic alignments were extracted from previously-calculated whole-genome alignments (UCSC).

## Homology-detection

To query for lncORF homologies, we developed a novel pipeline, GENOR, which integrates the main elements of our laboratory's previously-published efforts with manually-curated searches for homologues. Per ORF, GENOR conducts a forward jackhammer search against an in silico translated ORFs; the in silico translated smORF set from all NCBI-deposited ESTs (as of 02/2020, between 10 and 150 AA) was used for 12 different *Drosophila* species: *Drosophila simulans* (Dsim), *Drosophila sechelia* (Dsec), *Drosophila erecta* (Dere), *Drosophila yakuba* (Dyak), *Drosophila ananassae* (Dana), *Drosophila pseudoobscura* (Dpse), *Drosophila persimilis* (Dper), *Drosophila willistoni* (Dwil), *Drosophila mojavensis* (Dmoj), *Drosophila virilis* (Dvir) and *Drosophila grimshawii* (Dgri). All significant forward hits per species are evaluated in in reciprocal jackhammer searches against the whole ORFome of *Drosophila melanogaster*. If a forward hit is also the top reciprocal hit, the ORF is taken forward for further alignment evaluation using MAFFT[71] as well as custom scoring. Positions with fully-conserved residues ('*') were given a relative weight of 100; aligned amino acids with very similar biochemical properties (':'- scoring >0.5 in the PAM 250 matrix) were given a weight of "70", whereas residues with weak similarities ('.') were given a relative weight of 30. Per alignment, scores were added across positions, and divided by total query length to obtain a GENOR score for the hit.

## Statistics and reproducibility

**Representation factor**. The representation factor represents the probability that the observed overlap between two lists of ORFs could occur as often by random. The representation factor is calculated as the ratio of observed/expected overlaps, where the expected number of overlapping genes is obtained by multiplying the total number of genes in both groups, and dividing it by the total number of genes initially analysed[72]. A factor of 1 indicates that the overlap is that which is expected by random. A factor >1 indicates that overlap is higher than that expected by chance, whereas a representation factor <1 indicates less overlap than expected.

**Reproducibility**. Each experiment featured in Figs. 1f, g (lower panel on the latter); 3e; 4e; 5g have been repeated at least twice, with the same outcomes.

## Reporting summary

Further information on research design is available in the Nature Research Reporting Summary linked to this article.

# Data availability

All data generated in this study have been deposited in the NCBI's Gene Expression Omnibus with the GEO Series accession number GSE204739. Source data are provided with this paper.

# Code availability

The source code for GENOR is available at https://github.com/CouLab/GENOR.

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

## Acknowledgements

We thank Ana Hervas, Casimiro Baena and Eugénio Mangas for helpful discussions in the Lab, as well as Sarah Bishop, Ariel Bazzini, and Fernando Casares for comments on the manuscript. This work was funded by the following grants: PID2019-106227GB-I00 by the *Ministerio de Ciencia e Innovación/Agencia Estatal de Investigación* and CEX2020-00108-M *Unidad de Excelencia María de Maeztu* (J.P.C.); and by a Postdoctoral Fellowship DOC_01144/2020, awarded by the *Junta de Andalucía/CSIC* (P.P.).

## Author contributions

P.P., E.G.M., J.I.P. and A.P. designed and conducted the experiments; P.P. and J.P.C. conducted subsequent data analysis. P.P., E.G.M. and J.P.C. wrote the first draft. All authors contributed to the final manuscript.

## Competing interests

The authors declare no competing interests.
