## [Peer Review File · Nature Communications]

Translation and Natural Selection of micropeptides from long non-canonical RNAsREVIEWER COMMENTS

Reviewer #1 (Remarks to the Author):

The authors address the topic of “micro”-peptides encoded by genes annotated as long non-coding RNAs. This topic has been explored for over a decade now, with many studies and tools developed, but there still isn't a clear consensus in the field, so the topic remains timely. Before addressing the specific paper, let me summarize the current view in the field (in my opinion) about whether many lncRNAs are actually misannotated protein-coding genes. Multiple tools using different features for detection of evidence for production of functional proteins have been developed with classifying bona fide lncRNAs away from unannotated, and occasionally short, proteins, such as CPAT, CPC, PhyloCSF, RNCcode and many others. In parallel, large scale studies using Ribo-seq in mammalian tissues, such as the heart, have shown that in many lncRNAs, including some with well-established nuclear functions, there is evidence for translation of ORFs, that tend to be canonical, have Kozak sequences around their start codons, and as expected from cap-based translation initiation are biased to the 5' of the transcript. Also as expected, the translation of the RNAs that are more nuclear was shown to be less efficient, with an overall gradient of translation efficiencies (TEs), and with some ORFs only evident as translated only upon very deep sequencing. This is all consistent with the widespread occurrence of “translational noise”, i.e, bona fide translation events that are likely unavoidable on RNAs that reach the cytoplasm, but that do not necessarily result in stable and functional proteins, but rather produce rapidly degraded and ‘tolerated’ proteins, closely resembling the ones produced from uORFs. Notably, since mRNAs are typically more abundant than lncRNAs (~99% of polyadenylated molecules in a cell are mRNAs and not lncRNAs), uORFs contribute much more to these translation events than lncRNAs. While some proteomic evidence was found, typically for a small subset of these ORFs, there remain very few examples where such micropeptides were shown to be functional. Notably, virtually all these examples are supported by strong PhyloCSF or RNCcode scores, as these proteins are also highly conserved in evolution.

And so the current view is that the existing state-of-the-art tools that are used to sift through transcriptomes to identify lncRNAs, and that researchers studying specific lncRNAs are using before diving into functional experiments, are good enough to flag transcripts that are producing functional proteins (in contrast to transcripts that are just being translated to some extent but do not yield stable or functional proteins).

The current manuscript proposes to challenge this view and to conclude that a major fraction of the lncRNAs that they examine as annotated in *Drosophila* do encode proteins. It is well written, and the computational analyses and data collected are of overall high quality. I would argue that the presented data are actually not sufficient to challenge this view, which limits the impact of the manuscript. Most of the presented results that the authors elaborate on, such as better Kozak sequences driving more translation, 5' bias of translated ORFs, some transcripts (presumably abundant and cytoplasmic) being better translated etc., are all expected given our understanding of translation and previous studies in mammals. In any case, the analysis and data should match the current state-of-the art.

Major comments:

1. It is well known that the current sets of lncRNAs annotated in various databases such as RefSeq or Flybase were annotated long time ago and typically not using the currently available coding-potential predictors. The authors need to describe what the FlyBase lncRNA set they use is based on, run the commonly used tools (CPC, CPAT, PhyloCSF to name a few) and report how many of the lncRNAs pass these filters, and what is the correspondence between the lncRNAs that don't and the lncRNAs that they see as translated, or more importantly, translated and with evidence for ORF conservation in evolution.
2. The translation efficiency (estimated by the ratio of the Ribo-seq and RNA-seq reads) should be measured and compared for the lncRNA-derived ORFs, uORFs in 5'UTRs and canonical ORFs. This can be used to further evaluate how many of the ORFs that have some translation evidence are actually

likely to give rise to a substantial number of proteins, in particular when considering lncRNAs with reported noncoding functions.

3. I don't think the authors actually show that (from end of Introduction): "our findings support a model whereby lncRNAs acquire protein-coding function as their lncORFs evolve from an intermediate ribo-bound-only state to a fully translated one.". They show that there is some evidence of conservation in some lncRNA ORFs across closely-related species (again, unclear how much of this is just because of poor lncRNA annotations used). How does this prove some evolutionary dynamics? There is no evidence that genes that now have well-translated ORFs used to be only ribo-bound in the beginning, unless I missed something. The text should be rephrased accordingly.

4. In order to conclude (line 197) that "These results corroborate that extensive translational regulation is a general feature of lncRNA translation", the authors need to compare the specificity of lncRNA translation (examined as both Ribo-seq coverage and TE) when compared to protein-coding genes with matching expression levels. As presented, it is not clear if the enhanced specificity merely reflects lower expression levels, which would lead to more noise.

5. As mentioned, the evolutionary analysis of the translated ORFs is in my opinion the more interesting part of the paper. But it should be improved. First, the authors should identify and report separately the ORFs that have PhyloCSF- or RNaCode-based evidence of translation. These I would not call "lncRNAs", as these tools are commonly used to filter lncRNAs. For others, the authors should clearly present how many have ORFs with evidence of conservation vs. how many are expected by chance – what is the signal and what is the background, and hence what is the signal-background, i.e., how many lncRNAs they believe have evidence of conservation (both less mutations and high dN/dS scores) at different evolutionary depths over what is expected by chance. This is arguably the most interesting number – if a researcher now goes in to study a *Drosophila* lncRNAs that is noncoding based on PhyloCSF/RNaCode – what are the chances that this lncRNA has an ORF that is under selection, at least between closely related *Drosophila* species? Otherwise, the presented analysis just shows that there is some conservation signal in the lncRNAs as a whole, which is not surprising for an unfiltered set of lncRNAs, but that the bulk of the 1/3 of lncRNAs where they see translation evidence most likely produce non-functional and possible unstable, tolerated, protein products, unless proven otherwise.

Minor comments:

1. Line 53, it is not clear what "devoid of canonical ORFs" means for lncRNAs. Clearly lncRNAs have many ORFs, including many with great Kozak motifs, this is unavoidable in long RNA sequences.
2. The authors should show a metagene plot (and a heatmap, e.g., from deepTools) of Ribo-seq reads in the translated ORFs, to give a feeling to the extent that the start and end of translation events they annotate are correct (i.e., how often Ribo-seq coverage begins at the smORF start and ends at the smORF end?).

Reviewer #2 (Remarks to the Author):

In this manuscript, Patraquim et al explore the translation of lncRNAs in *D. melanogaster* at different developmental stages. I think by now even the strongest sceptics of lncRNA translation agree that at least some lncRNAs are translated at least to some extent. From this point of view Patraquim et al findings aren't novel. What is novel is certain methodological innovations in the analysis of lncRNAs translation and evolution of translated ORFs. Specifically, they were able to implement a triplet periodicity signal in the detection of translated ORFs, previously detection of ORF translation was based only on coverage or read length distribution, as low expression of lncRNAs made it difficult to detect the periodicity. Due to these methodological advances, the authors were able to characterise lncRNA translation with higher accuracy. This allowed them to characterise the features of lncRNA ORFs associated with their high translation. These features suggest the mechanistic explanation of this translation which is consistent with the predominant leaky scanning mode of translation initiation. By using an HMM-based approach, the authors were able to detect homologous lncRNA ORFs that cannot

be normally detected due to their short length and low sequence similarity. This enabled phylogenetic analysis of translated ORFs which provided evidence of purifying selection on at least some of these ORFs providing strong evidence for their functionality.

Critical comments

1. Terminology

In the title/abstract the authors suggested renaming lncRNAs from 'Long Non-Coding RNAs' to 'Long Non-Canonical mRNAs'. I am fully supportive of the authors' desire to change the term. Too often in Biology terms are introduced well before we start understanding the subjects of these terms. Thus, we end up with oxymoron terms and lncRNAs is a good example. I also admire the clever idea of retaining the same abbreviation. However, I do not see it as a considerable improvement as it may bring additional confusion. If there are long non-canonical RNAs, there should be long canonical RNAs. What are they? Presumably, that should be mRNAs. But as mRNAs are not called "canonical RNAs", the term non-canonical RNAs is unclear. Perhaps the authors could go a step further and term them 'long non-canonical mRNAs' (lncmRNAs or lncRNAs if m is dropped in the abbreviation)? While such term change is more confrontational it is clearer and I think that is what the authors meant but stopped halfway towards it. However, 'long' is still a problem as there are no shorter canonical or non-canonical mRNAs. The issue is not simple and while I agree that it would be nice to rename lncRNAs, this should be done carefully and a new term should be introduced only if it is a clear improvement. The terminological issue becomes even more problematic in the main manuscript text when smORFs in lncRNAs are abbreviated as lncORFs.

"Long non-canonical RNA micropeptides" in the title also doesn't work for two reasons: 1. RNA peptides? 2. Long micropeptides? Perhaps something like "Translation of *D. melanogaster* lncRNAs and natural selection of microproteins encoded in them" would be a more clear and specific title for this manuscript? Or, if Nature Communication does not allow abbreviations: "Evolution and synthesis of non-mRNA encoded microproteins in *D. melanogaster*". It would be nice to avoid "translation of microproteins" as strictly speaking it is RNA that is translated.

2. Differential translation.

A potentially interesting finding of the manuscript is that translation of lncRNA smORFs varies more across the developmental stages than that of protein coding ORFs. However, it is not clear to me to what extent this observation could be simply due to higher variability in detection of smORF translation because of their low coverage (Fig1B). For lowly translated ORFs the stochastic differences in ribosome profiling signal may push smORFs below or above the detection thresholds. Also, I am puzzled by the sentence: "Only 17% of differentially translated lncORFs between different stages show significant changes in TE (Fig.2d), while the rest show qualitatively stable ribosomal binding, indicating that this translational regulation does not result from general quantitative changes in ribosomal binding to lncRNAs." Isn't it self-contradictory? My understanding is that differential translation means significant changes in TE. If not, how do you define differential translation?

3. Features associated with robust lncRNA ORF translation.

In the identification of the features associated with the translation of ORFs, the authors used riboseq coverage as one of the features (Figure 4a), but in my understanding this was used as one of the parameters for the detection of translated ORFs (Methods section "Defining a set of bona-fide translated ORFs"). The periodicity signal is also expected to correlate with the coverage. Does it even make sense to do such an analysis? The dependence is expected by the design. Regarding the Kozak score, I wonder if a sequence logo of Kozak contexts (excluding AUG) for individual classes would be useful in Figure 4B. The rank preference of ORFs positions (Figure 4c) makes a lot of sense from the mechanistic point of view. But I wonder how accurate this analysis is in the absence of the data on transcription starts. CAGE-seq data could be really useful here. RNA-seq alone may not be sufficient for the situations where there is more than one transcription start. For example, in the case of the human MTLN gene (formerly lncRNA LINC00116), the protein coding region is located at the end of

the long MTLN RNA isoform. However, Zaheed et al (doi: 10.3389/fcell.2021.703374) recently argued that the protein coding ORF is translated only in the short RNA isoforms where it is indeed close to the 5' end. Thus, I suspect that the provision of transcription start sites could actually strengthen the authors' conclusion.

4. Lack of analysis of the features associated with robust lncRNA translation (as opposed to individual ORFs).

The authors analyzed the features associated with the robust translation of ORFs and these features, as expected, are strongly associated with efficient translation initiation. This is likely because more preinitiation complexes would initiate at the corresponding ORFs providing a stronger riboseq signal (coverage + periodicity). But is there a difference in the ribosome engagement with lncRNAs in the first place, i.e. how efficient preinitiation complexes are assembled at 5' caps of lncRNAs? To me, this is a very interesting and important question, which, unfortunately, was not addressed by the authors.

5. ORFs are limited to AUG only.

I wonder whether the analysis would be more comprehensive if non-AUG initiated ORFs would also be explored. Low efficient translation initiation at non-AUG codons is widespread in mammals but has also been reported in lncRNAs of *D. melanogaster* (Montigny et al doi: 10.1186/s13059-021-02345-8).

Minor comments

Line 37 "Thus smORFs sit at the interface between coding and non-coding RNA" the use of "interface" does not seem appropriate here, my understanding is that smORFs simply occur in all RNAs.

Fig.1a, no units on X and Y axes (RPKM?); 1c - font size of y axis ticks and legend is non legible

Line 162-163, 'lncORFs with - framing corroborated in multiple RPF sizes) display strong and reproducible FLAG signs'; it's not clear what is '-' framing; extra parenthesis

Fig.3F, colours on legend and plot are not matched ('Ribo-bound' category is white, while on plot it's pink?), x axis labels are absent

Methods section. I understand that for brevity almost every step points to another study - Patraquimet al. (2020). Would it be possible to add some brief explanation into the main text as well? E.g. lines 644-646, which ORF classes were used to remove overlaps? Only CDSs or uORFs/dORFs?

Methods, lines: 715-716, when calculating log odds ratio between observed and background nucleotide frequencies for CDSs starts, what was taken as background nucleotides frequencies?

Methods, linesL 721, extra parenthesis

Supplementary figure 3c, the gradient colour bar is empty

Reviewer #3 (Remarks to the Author):

The manuscript by Patraquim and colleagues describes the translation of a set of long non-coding RNAs in *Drosophila melanogaster*. By generating extensive ribosome profiling during three embryogenesis stages, the authors extended the findings of their work published in 2020 and convincingly found translation of a few hundreds of lncRNAs and determined several lncRNA features that favor translation at the gene level. Moreover, they propose to change the name of these genes to

'long non-canonical RNAs'.

Although some of the conclusions of the paper about lncRNA translation and evolution have been already studied in other species (e.g. human, mouse, yeast), this topic is rather unexplored in *Drosophila* yet, hence the authors made an original contribution to the field. The work partially supports the main conclusions of the paper, but some additional evidence and details should be included before publication (see major comments below).

Major comments:

Finding homologues for microproteins encoded by smORFs is a hard task and different approaches have been proposed in the last years. One limitation is the selection of query databases to search for homologues. Here the authors use transcriptomes and ESTs from 12 different species. For how many of the species are the transcriptomes/ESTs corresponding to matched embryogenesis stages? Otherwise, ORFs/transcripts which are uniquely expressed at these stages might remain undetected in the transcriptomes of other species if they are not expressed in the conditions used to generate the transcriptome/ESTs.

Also, the authors used a GENOR cut-off of 50 to detect homologues but did not explain how this score was selected. If this was an arbitrary cut-off, using alternative cut-offs (for instance 30 and 70) would help to determine if the results are robust or are quite variable depending on the threshold. How is the score calculated if the ORF is significantly shorter in another species? For instance, if one ORF is 50AA in one species and aligns to another ORF which is 30AA because it has a premature stop codon, is the score calculated over the 30 aligned positions?

The authors suggest that many ORFs evolved de novo, but I didn't find a proper analysis for this specific mechanism of evolution besides running GENOR. Do the authors assume that all ORFs aligned to regions with GENOR < 50 emerged de novo? There might be still several ORFs that only aligned to regions with score > 50, and these cases would fit the definition of 'orphan' but not necessarily emerged de novo. Moreover, did the authors search for homology in the same species transcriptome/proteome -without an upper length cut-off? Some of these ORFs might be pseudogenized or (partially) duplicated regions, but as the searches were limited to ORFs in other species and in the range of 10-150AA, then these alignments would not be identified. These are just some ideas to determine how many of the ORFs seem to have evolved de novo in contrast to alternative mechanisms of evolution.

Finally, I miss some main table(s) with all the information collected in the paper, such as ORF coordinates and sequences, ORF categories, translated lncRNAs, and evolutionary information.

Minor comments:

Line 14: "Apparently unable to produce peptides, lncRNA function seems to only involve RNA sequence and structure." lncRNA function can also be associated with expression (regardless of sequence or structure), so I would suggest adding this possibility as well.

Line 22: "Our results expand the repertoire of lncRNA functions" Even though the authors find some evidence of stage-specific translation, protein production, and selection, they did not perform any functional analysis so this claim should be toned down.

Line 50: "not only regulating the translation of the 50 canonical protein located downstream, but also producing short peptides (~25 AA) that can interact with it" While this is true for some cases -for instance, some examples given by Chen et al.-, uORFs can also produce proteins whose functions are independent of the main protein, so it would be good to highlight this possibility (e.g. PMID: 33406399 or PMID: 33449506).

Line 108: Are transcribed lncRNAs the ones that had at least one ORF with RPKMRNA > 1?

Line 154: Are there ORF length differences between the different ORF states? It would be interesting to see this in a plot.

Line 156: I am a bit confused about what is a 'translation event'. Please elaborate.

Line 167: IRIN should point to fig. 1G

Line 203: Are translated lncRNAs more highly expressed? For comparison, what % of protein-coding genes are translated? This is maybe described in the author's previous paper, but I miss this information in this manuscript.

Line 322: The authors should show the distribution of ORF lengths per degree of conservation, separated by ORF stages.

Line 399: If ribosome profiling datasets of other tissues and conditions in *Drosophila melanogaster* are publicly available, it would be interesting to see how many of the 'ribo-bound ORFs' remain in the same state or if, otherwise, are highly translated at specific conditions or tissues.

Line 413: "In turn, this purifying selection is strong proof that the encoded micropeptides are translated and functional." The ORFs were compared as a group so there is no individual evidence of possible selection and functionality. Hence, the sentence should refer to the enrichment in this group rather than considering that all the micropeptides are selected.

Line 441: "However, unicellular eukaryote genomes include uORFs but not lncRNAs,". This statement is not correct. Unicellular eukaryote genomes also contain lncRNAs, although at lower numbers. Due to the small size of these genomes, most of these transcripts overlap protein-coding genes in antisense configurations. Since the authors are not only describing lincRNAs, antisense genes would be in principle considered as lncRNAs. It would be good to know how many of the expressed and translated lncRNAs in *Drosophila* are antisense genes.

Line 453: "Similarly, micropeptides produced by a lncRNA have been shown to enhance the function of miRNAs produced by the same lncRNA." How many lncRNAs which act as small RNA hosts are translated? Is there an enrichment of this RNA category in the set of translated RNAs?

Line 641: What is the upper length cut-off? 150 or 450 codons? Did you see any lncRNA with translated ORFs above this length? Maybe a small % of lncRNAs that the authors don't find as translated contain longer ORFs, probably some unannotated protein-coding genes or pseudogenes.

Line 643: This methods section is difficult to understand. Does this line mean that any internal ORF overlapping a longer one was removed? Does it refer to the same frame or any of the three possible frames? What are duplicates here? How was the set of 18,507 ORFs reduced to 16,335?

Line 727: What is the minimum number of substitutions required to calculate dN/dS? The authors should make this information available in a table, including the total number of substitutions.

REVIEWER COMMENTS

Reviewer #1 (Remarks to the Author):

We agree with most of the general comments of this reviewer, but we would like to point out that our bioinformatic and genomic analyses are always in feed-back with our functional studies (more recently in Magny et al. 2021, Nat Comm). However, we do not agree that “very few examples where such micropeptides were shown to be functional”, rather, that in very few cases such micropeptides were fully characterised and their function ascertained at the genetic and molecular levels. Also, it would be interesting to re-evaluate the mentioned bioinformatic and genomic studies in vertebrates with our tools.

Major comments:

1. It is well known that the current sets of lncRNAs annotated in various databases such as RefSeq or Flybase were annotated long time ago and typically not using the currently available coding-potential predictors. The authors need to describe what the FlyBase lncRNA set they use is based on, run the commonly used tools (CPC, CPAT, PhyloCSF to name a few) and report how many of the lncRNAs pass these filters, and what is the correspondence between the lncRNAs that don't and the lncRNAs that they see as translated, or more importantly, translated and with evidence for ORF conservation in evolution.

As an initial lncRNA set, we use noncoding RNA transcripts from Flybase release 6.13 (dmel-all-ncRNA-r6.13.fasta) as an input for getORF, an ORF predictor (**options:** -find 1 -reverse No -sequence). The minimum size is 10 AA and there is no maximum ORF size. This Flybase lncRNA annotation is based on different gene prediction methods, such as Augustus (Stanke, et al. 2006, BMC Bioinformatics 7:62), Contrast (Gross, Do, and Batzoglou, 2005, BCATS 2005 Symposium Proceedings, p. 82), GeneID (Parra, Blanco, and Guigo, 2000, Genome Research, 10: 511-515), NCBI Gnomon (Suvorov, et al. 2006), and SNAP. As the reviewer points out, these methods have been revealed (by previous papers of ours and others) as probably ill-suited to assess the coding potential of short sequences. A major issue is that even the most up to date predictors also struggle with short sequences; but we have to start somewhere, and our starting point is the genome annotation. Testing the “codingness” of lncRNAs with more appropriate methods is what we do with GENOR and dN/dS, and this is a result of the paper, not its starting point. It is not our fault that thousands of RNAs were prematurely assigned to a functional class (“non coding”); we want to be part of the solution, not part of the problem.

However, as the reviewer suggested, we have now ran *Drosophila melanogaster* lncRNAs through PhyloCSF and AnAblast (Lin et al. 2011; Rubio et al. 2019), two current tools to predict coding potential, and still found very little overlap with our Ribo-Seq based results (resembling false positive rates, (FigS1F)). We suspect PhyloCSF fails because it interprets any mis-alignment or synteny loss as “non-coding”, but we haven't explored the reasons why these or Flybase tools don't work on our experimentally-validated dataset, since the bioinformatic tools we present (GENOR,

dN/dS) offer a much better match between translation and conservation. Altogether, these results show that experimental assessment of translation is essential to assess translation of short sequences.

2. The translation efficiency (estimated by the ratio of the Ribo-seq and RNA-seq reads) should be measured and compared for the lncRNA-derived ORFs, uORFs in 5'UTRs and canonical ORFs. This can be used to further evaluate how many of the ORFs that have some translation evidence are actually likely to give rise to a substantial number of proteins, in particular when considering lncRNAs with reported noncoding functions.

We have now added the TE values for canonical ORFs and uORFs to a new version of Fig. 4A. However, it is important to note that although higher TE correlates with translation, on its own (like Kozak or other factors we identify) is not a good predictor of translation for individual ORFs. Also, TE not a measure of total peptide produced; $RPKM^{FP}$ is, but $RPKM^{FP}$ is low on average for lncRNAs, and although we use a filter of >1 , as used for canonicals, we also identify a class of “Ribo-bound-only” uORFs (Patraquim 2020) and lncORFs (this work) that display good $RPKM^{FP}$, yet no apparent translation: as seen here and elsewhere (Ingolia 2009), framing, is by far the best standalone indicator of translation.

3. I don't think the authors actually show that (from end of Introduction): “our findings support a model whereby lncRNAs acquire protein-coding function as their lncORFs evolve from an intermediate ribo-bound-only state to a fully translated one.”. They show that there is some evidence of conservation in some lncRNA ORFs across closely-related species (again, unclear how much of this is just because of poor lncRNA annotations used). How does this prove some evolutionary dynamics? There is no evidence that genes that now have well-translated ORFs used to be only ribo-bound in the beginning, unless I missed something. The text should be rephrased accordingly.

We have rephrased this sentence, it now reads: “our findings are compatible with the possibility that lncRNAs acquire protein-coding function as their lncORFs evolve from an intermediate ribo-bound-only state to a fully translated one.”

4. In order to conclude (line 197) that “These results corroborate that extensive translational regulation is a general feature of lncRNA translation”, the authors need to compare the specificity of lncRNA translation (examined as both Ribo-seq coverage and TE) when compared to protein-coding genes with matching expression levels. As presented, it is not clear if the enhanced specificity merely reflects lower expression levels, which would lead to more noise.

The reviewer makes a good point, and we have now added this analysis in Fig. S2, and re-written the section. Using the 300-400 canonical ORFs with $RPKM^{FP}$ within the observed lower 75% of translated lncORFs ($1 < RPKM < 7.5$), we observe limited levels of translational regulation, levels that still resemble those for all canonicals (Patraquim 2020). This control indicates that low expression levels do not artifactually introduce the observed translational regulation in lncORFs. Also, note that we use Z-scores

(already a normalised parameter) ratios to define significant changes in translation, i.e. changes beyond the standard deviation of the studied ORF sample. Another part of the regulation seems to come from qualitative changes in framing, and is worth remembering that our binomial pipeline is RPKM-independent (Patraquim 2020).

5. As mentioned, the evolutionary analysis of the translated ORFs is in my opinion the more interesting part of the paper. But it should be improved.

-First, the authors should identify and report separately the ORFs that have PhyloCSF- or RNaCode-based evidence of translation. These I would not call “lncRNAs”, as these tools are commonly used to filter lncRNAs.

See discussion of point 1, only very few lncORFs with good PhyloCSF scores have Ribo-Seq based evidence of translation. We did not name these RNAs as non-coding; if we are revising an annotation, it follows that we must assess the annotation itself, not our attempted (and failed) improvement of the annotation. Still, we have added the PhyloCSF results (Fig. S1).

For others, the authors should:

- clearly present how many have ORFs with evidence of conservation vs. how many are expected by chance – what is the signal and what is the background, and hence what is the signal-background, i.e., how many lncRNAs they believe have evidence of conservation (both less mutations and high dN/dS scores) at different evolutionary depths over what is expected by chance. This is arguably the most interesting number – if a researcher now goes in to study a *Drosophila* lncRNAs that is noncoding based on PhyloCSF/RNaCode – what are the chances that this lncRNA has an ORF that is under selection, at least between closely related *Drosophila* species?

Otherwise, the presented analysis just shows that there is some conservation signal in the lncRNAs as a whole, which is not surprising for an unfiltered set of lncRNAs, but that the bulk of the 1/3 of lncRNAs where they see translation evidence most likely produce non-functional and possibly unstable, tolerated, protein products, unless proven otherwise.

Unfortunately, we do not have enough data for different evolutionary depths, due to the limitations of available data. To generate better transcriptomes and Riboseq data in other species, we would need a dedicated research project and another paper.

The reviewer seems to have missed the supplementary file depicting conservation depth and dN/dS for each lncORF. To avoid similar issues for future readers we have now called that file in the text, together with the relevant figures.

We do find it surprising that translated lncORFs show conservation signal, since Orera et al. 2016 only found evidence of neutral evolution.

On a more general level, the reviewer seems concerned that random translational and conservation noise could be interpreted as evidence of translation and function. Indeed, any individual dN/dS result can be a false positive, the key is the average values and the total number of positives. We note:

- 1- For dN/dS, Ribo-bound-only IncORFs serve as control; they display dN/dS near 1, typical of non-coding sequences. The calculation we use requires a minimum of 2 substitutions (1 non syn, 1 syn to avoid zeroes and infinities), there are IncORFs for which dN/dS cannot be computed as they have 0 synonymous or non-synonymous substitutions.
- 2- It is important that dN/dS data is independently generated from available genomic micro-alignments (UCSC, multiz alignments, see methods), then assorted by us to translated/non-translated pools. Thus, it would be very difficult to explain why two independent samples (translated and dN/dS<1) should overlap significantly beyond what is expected by random chance (we have calculated the representation factor of this overlap to measure this and included it in the main text and methods).

Minor comments:

1. Line 53, it is not clear what “devoid of canonical ORFs” means for IncRNAs. Clearly IncRNAs have many ORFs, including many with great Kozak motifs, this is unavoidable in long RNA sequences.

We have rephrased this to: “devoid of canonical, **annotated** ORFs”. Indeed, we showed in Couso and Patraquim (2017) that most IncORFs seem to be generated at random, but we are not aware of good Kozak motifs appearing by chance in the numbers we observe amongst non-coding sequences. In addition, the Kozak motifs we report here correlate with translation status.

On balance, it is extremely unlikely that a collection of independent variables will display significant overlap, in other words, that somehow we have obtained a “random” sample of actually non-translated IncORFs, which also happen to show statistically significant differences for all of: framing, TE, dN/dS, Kozak, ORF length (see point 4 of Rev. 2) and Cistronicity.

2. The authors should show a metagene plot (and a heatmap, e.g., from deeptools) of Ribo-seq reads in the translated ORFs, to give a feeling to the extent that the start and end of translation events they annotate are correct (i.e., how often Ribo-seq coverage begins at the smORF start and ends at the smORF end?).

This is a good suggestion and a standard in the field, we had removed it to lighten the manuscript. We have now included this plot in Fig. S1a (right panel).

Reviewer #2 (Remarks to the Author):

We thank this reviewer for his/her general comments on our work.

Critical comments

1. Terminology

In the title/abstract the authors suggested renaming lncRNAs from 'Long Non-Coding RNAs' to 'Long Non-Canonical mRNAs'. I am fully supportive of the authors' desire to change the term. Too often in Biology terms are introduced well before we start understanding the subjects of these terms. Thus, we end up with oxymoron terms and lncRNAs is a good example. I also admire the clever idea of retaining the same abbreviation. However, I do not see it as a considerable improvement as it may bring additional confusion. If there are long non-canonical RNAs, there should be long canonical RNAs. What are they? Presumably, that should be mRNAs. But as mRNAs are not called "canonical RNAs", the term non-canonical RNAs is unclear. Perhaps the authors could go a step further and term them 'long non-canonical mRNAs' (lncmRNAs or lncRNAs if m is dropped in the abbreviation)? While such term change is more confrontational it is clearer and I think that is what the authors meant but stopped halfway towards it. However, 'long' is still a problem as there are no shorter canonical or non-canonical mRNAs. The issue is not simple and while I agree that it would be nice to rename lncRNAs, this should be done carefully and a new term should be introduced only if it is a clear improvement. The terminological issue becomes even more problematic in the main manuscript text when smORFs in lncRNAs are abbreviated as lncORFs.

"Long non-canonical RNA micropeptides" in the title also doesn't work for two reasons: 1. RNA peptides? 2. Long micropeptides? Perhaps something like "Translation of *D. melanogaster* lncRNAs and natural selection of microproteins encoded in them" would be a more clear and specific title for this manuscript? Or, if Nature Communication does not allow abbreviations: "Evolution and synthesis of non-mRNA encoded microproteins in *D. melanogaster*". It would be nice to avoid "translation of microproteins" as strictly speaking it is RNA that is translated.

We agree with the reviewer on the unfortunate nomenclature we find ourselves with (stemming from sweeping and premature decisions in the 1990's/2000's). However, some of our nomenclature was already proposed in 2017 by us and has now been incorporated into a community white paper in press in Nature Biotechnology.

Regarding the title, the absence of "lncRNA" would rob the paper of some of its intended impact – a direct critique of the "lncRNA" concept as opposed and mutually exclusive to coding mRNAs. However, we agree that our title could induce confusion (we just decry the character limit on Nat Comm. Coupled with the shortage of declensions in English). We have changed it to "Translation and conservation of micropeptides from long-non-canonical-RNAs" using anti-aesthetic hyphens to clarify that "long-non-canonical" are adjectives referring to the noun "RNAs".

Notice that we are not trying to introduce a new RNA class, but to stimulate a re-evaluation of the *entire* lncRNA class. Indeed, lncRNAs that appear not translated in our samples could be translated in other developmental stages, or under different environmental conditions, or in other species. The term “Long-non-canonical-RNAs” does not prejudice the function of these RNAs, yet maintain their distinction (for the time being) from others whose function is clearer (mRNAs, rRNAs, miRNAs, snoRNAs, etc.). We could do an analogy with viral RNA: it can act as mRNA, but also as genomic RNA; it is not just viral mRNA, but something else.

2. Differential translation.

A potentially interesting finding of the manuscript is that translation of lncRNA smORFs varies more across the developmental stages than that of protein coding ORFs. However, it is not clear to me to what extent this observation could be simply due to higher variability in detection of smORF translation because of their low coverage (Fig1B). For lowly translated ORFs the stochastic differences in ribosome profiling signal may push smORFs below or above the detection thresholds.

See point 4 of Rev.1, we have re-written this section for clarity and added a control with canonical lowly expressed ORFs in Fig. S2 that shows how translational regulation in lncORFs is not due to low expression levels artefacts.

Also, I am puzzled by the sentence: “Only 17% of differentially translated lncORFs between different stages show significant changes in TE (Fig.2d), while the rest show qualitatively stable ribosomal binding, indicating that this translational regulation does not result from general quantitative changes in ribosomal binding to lncRNAs.” Isn't it self-contradictory? My understanding is that differential translation means significant changes in TE. If not, how do you define differential translation?

Sorry, this was a typo! It should have been written “while the rest show *quantitatively* stable ribosomal binding”. In the new text we distinguish between qualitative changes (framing) and quantitative ones (TE).

3. Features associated with robust lncRNA ORF translation.

In the identification of the features associated with the translation of ORFs, the authors used riboseq coverage as one of the features (Figure 4a), but in my understanding this was used as one of the parameters for the detection of translated ORFs (Methods section “Defining a set of bona-fide translated ORFs”). The periodicity signal is also expected to correlate with the coverage. Does it even make sense to do such an analysis? The dependence is expected by the design.

Indeed the filter used for detection of ribosome-bound ORFs was RPKM >1 in both replicates in a given stage, but we are not sure if this could bias the whole sample. Indeed, only a subset of these RPKM>1 ribo-bound ORFs display framing, with most lncORFs with RiboSeq RPKM>1 showing no periodicity. However, we agree that this correlation may appear disputable, so we have removed RPKM as a marker for translation and substituted it with TE. TE was not part of the pipeline filters, and while an average of TE=1 might be expected (since the filters were RPKMFP≥1 and

RPKMRNA \geq 1, average TE could be expected to hover near 1), the observed average TE for robustly translated IncORFs is 1.9, while for ribo-bound-only is 0.6.

Regarding the Kozak score, I wonder if a sequence logo of Kozak contexts (excluding AUG) for individual classes would be useful in Figure 4B.

This is a good suggestion and a more standard way to present these results, we have therefore added the sequence logos and substituted them for the original panel in Fig. 4, which is now in Fig. S4. Interestingly, a qualitative difference in base composition in position 4 can now be noted between translated and ribo-bound thanks to the logos, which likely impinges on framing and translation.

The rank preference of ORFs positions (Figure 4c) makes a lot of sense from the mechanistic point of view. But I wonder how accurate this analysis is in the absence of the data on transcription starts. CAGE-seq data could be really useful here. RNA-seq alone may not be sufficient for the situations where there is more than one transcription start. For example, in the case of the human MTLN gene (formerly IncRNA LINC00116), the protein coding region is located at the end of the long MTLN RNA isoform. However, Zaheed et al (doi: 10.3389/fcell.2021.703374) recently argued that the protein coding ORF is translated only in the short RNA isoforms where it is indeed close to the 5' end. Thus, I suspect that the provision of transcription start sites could actually strengthen the authors' conclusion.

The flybase transcriptomes are exceedingly well annotated, and thus it is unlikely that transcriptional start sites have been missed. We use the longest RNA isoform for our rank calculations, which works against the observed result, and would do so even if new start sites (and further upstream ORFs) existed, these would decrease the rank of all IncORF classes, such as the lower rank of translated IncORF would still prevail.

Importantly, regarding the existence of alternative isoforms, even if we obtained even more precise transcriptional start sites of alternative isoforms using CAGE-Seq, it would be impossible to determine from which isoform the Ribo-Seq footprints come from.

4. Lack of analysis of the features associated with robust IncRNA translation (as opposed to individual ORFs).

The authors analyzed the features associated with the robust translation of ORFs and these features, as expected, are strongly associated with efficient translation initiation. This is likely because more preinitiation complexes would initiate at the corresponding ORFs providing a stronger riboseq signal (coverage + periodicity). But is there a difference in the ribosome engagement with IncRNAs in the first place, i.e. how efficient preinitiation complexes are assembled at 5' caps of IncRNAs? To me, this is a very interesting and important question, which, unfortunately, was not addressed by the authors.

The reviewer makes an interesting point. But, this entails an entire research program and stand-alone paper. There are a number of features that could correlate with increased ribosomal engagement, but TE reflects all of these. We already show a

number of RNA features that correlate with translation: lncRNA RNA length, transcription level, polyA, cytoplasmic localisation (we explain the protocol further, clarifying that it begins with a cytoplasmic extraction), and now add, interestingly, exclusion of miRNA production (also responding to a comment from reviewer 3).

5. ORFs are limited to AUG only.

I wonder whether the analysis would be more comprehensive if non-AUG initiated ORFs would also be explored. Low efficient translation initiation at non-AUG codons is widespread in mammals but has also been reported in lncRNAs of *D. melanogaster* (Montigny et al doi: 10.1186/s13059-021-02345-8).

We agree that including non-AUG ORFs would bring further completeness to our analysis of lncRNA translation. However, we are reluctant to include these in our manuscript since compared to canonical AUG ORFs, non-AUG ORFs would represent a minority, and would bring an extra layer of controversy to the manuscript. Further, much of our analytical efforts were focused on maximizing the number of reads analysed to obtain reliable framing in *Drosophila* lncORFs, as the reviewer point out it is expected that non-AUG start codons would have even lower efficiencies of translational initiation, which dampen our ability to reliably detect framing on these ORFs. We are confident in our results for AUG ORFs, and feel it is unwise and besides the main biological question to expand at this point on non-AUG translation.

Minor comments

Line 37 “Thus smORFs sit at the interface between coding and non-coding RNA” the use of “interface” does not seem appropriate here, my understanding is that smORFs simply occur in all RNAs.

We have removed this sentence.

Fig.1a, no units on X and Y axes (RPKM?); 1c - font size of y axis ticks and legend is non legible

Corrected.

Line 162-163, ‘lncORFs with - framing corroborated in multiple RPF sizes) display strong and reproducible FLAG signs’; it’s not clear what is ‘-’ framing; extra parenthesis

We have corrected this typo.

Fig.3F, colours on legend and plot are not matched (‘Ribo-bound’ category is white, while on plot it’s pink?), x axis labels are absent

We have made these corrections to the relevant figures.

Methods section. I understand that for brevity almost every step points to another

study - Patraquimet al. (2020). Would it be possible to add some brief explanation into the main text as well? E.g. lines 644-646, which ORF classes were used to remove overlaps? Only CDSs or uORFs/dORFs?

We have added these precisions: “Finally, the coordinates of these ORFs were compared with other ORF classes (command *bedtools intersect -v -s -a*; no overlaps with both annotated coding sequences and uORFs) to obtain a set of 16,335 unique, non-overlapping IncORFs for further analysis.”

Methods, lines: 715-716, when calculating log odds ratio between observed and background nucleotide frequencies for CDSs starts, what was taken as background nucleotides frequencies?

This has been specified in methods, it now reads: “we then calculated the *log* odds ratio between observed and background nucleotide frequencies -the latter calculated as position-independent relative frequencies of the same sequences”

Methods, linesL 721, extra parenthesis

Ok, corrected

Supplementary figure 3c, the gradient colour bar is empty

ok, corrected

Reviewer #3 (Remarks to the Author):

The manuscript by Patraquim and colleagues describes the translation of a set of long non-coding RNAs in *Drosophila melanogaster*. By generating extensive ribosome profiling during three embryogenesis stages, the authors extended the findings of their work published in 2020 and convincingly found translation of a few hundreds of lncRNAs and determined several lncRNA features that favor translation at the gene level. Moreover, they propose to change the name of these genes to 'long non-canonical RNAs'.

Although some of the conclusions of the paper about lncRNA translation and evolution have been already studied in other species (e.g. human, mouse, yeast), this topic is rather unexplored in *Drosophila* yet, hence the authors made an original contribution to the field. The work partially supports the main conclusions of the paper, but some additional evidence and details should be included before publication (see major comments below).

Major comments:

1-Finding homologues for microproteins encoded by smORFs is a hard task and different approaches have been proposed in the last years. One limitation is the selection of query databases to search for homologues. Here the authors use transcriptomes and ESTs from 12 different species. For how many of the species are the transcriptomes/ESTs corresponding to matched embryogenesis stages? Otherwise, ORFs/transcripts which are uniquely expressed at these stages might remain undetected in the transcriptomes of other species if they are not expressed in the conditions used to generate the transcriptome/ESTs.

Since available transcriptomes for these species included very few lncRNAs, we resorted to use ESTs. Unfortunately, although we have used all available ESTs per species, sometimes stage information is not available for these. Although the total coverage is substantial (339.957 distinct ESTs) it is correct and likely that we may have missed a number of lncRNAs. This is one of the reasons why we could not extend our natural selection analysis beyond *D.simulans/D. sechelia*. However, this loss would be randomly distributed, and thus should not affect the main observation, i.e. the overlap between conservation and translation. Our study is not meant to be an exhaustive record of natural selection acting on lncORFs, but rather to establish this principle, and its general characteristics.

2-Also, the authors used a GENOR cut-off of 50 to detect homologues but did not explain how this score was selected. If this was an arbitrary cut-off, using alternative cut-offs (for instance 30 and 70) would help to determine if the results are robust or are quite variable depending on the threshold. How is the score calculated if the ORF is significantly shorter in another species? For instance, if one ORF is 50AA in one species and aligns to another ORF which is 30AA because it has a premature stop codon, is the score calculated over the 30 aligned positions?

The cut-off of 50 was determined empirically, based on our previous manual homology detection methods (Galindo et al. 2007, Magny et al 2013, Pueyo et al 2016, Magny et al 2021). The score is always calculated based on the length of the hit, so in the example of the reviewer, it would be the aligned 30 AA positions.

The authors suggest that many ORFs evolved *de novo*, but I didn't find a proper analysis for this specific mechanism of evolution besides running GENOR. Do the authors assume that all ORFs aligned to regions with GENOR < 50 emerged *de novo*?

Not at all. Further and specific work would be needed to back up such a statement. For clarity, we have re-worded this statement to "Overall our findings **are compatible with the possibility** that lncRNAs acquire protein-coding function as their lncORFs evolve from an intermediate ribo-bound-only state to a fully translated one." (see rev. 1, major comment 3).

There might be still several ORFs that only aligned to regions with score > 50, and these cases would fit the definition of 'orphan' but not necessarily emerged *de novo*. Moreover, did the authors search for homology in the same species transcriptome/proteome -without an upper length cut-off? Some of these ORFs might be pseudogenized or (partially) duplicated regions, but as the searches were limited to ORFs in other species and in the range of 10-150AA, then these alignments would not be identified. These are just some ideas to determine how many of the ORFs seem to have evolved *de novo* in contrast to alternative mechanisms of evolution.

We now add AnABlast results, which seem to discard that most translated and conserved lncORFs are pseudo-genes. AnABlast is an algorithm which detects canonical-like genomic sequences, and is especially effective at detecting pseudo-genes (Jimenez et al. 2015, DNA res. PMID: 26494834), we have now included these results in the manuscript. However, we changed discussion and Fig. 6B to 'novel' genes, which does not pretend to pinpoint the actual moment of gene origin.

Finally, I miss some main table(s) with all the information collected in the paper, such as ORF coordinates and sequences, ORF categories, translated lncRNAs, and evolutionary information.

We had provided a file with all the data in the initial submission, and are unsure as to why this was not made available to reviewers; we have made sure that they are better referenced in the text.

Minor comments:

Line 14: "Apparently unable to produce peptides, lncRNA function seems to only involve RNA sequence and structure." lncRNA function can also be associated with expression (regardless of sequence or structure), so I would suggest adding this possibility as well.

This is a fair point, we have reworded this statement to “apparently unable to produce peptides, lncRNA function seems to rely only on RNA expression, sequence or structure.”

Line 22: “Our results expand the repertoire of lncRNA functions” Even though the authors find some evidence of stage-specific translation, protein production, and selection, they did not perform any functional analysis so this claim should be toned down.

We refer to the coding, peptide production, capabilities of these genes in general, not whichever specific biological function of the peptides; we have therefore changed “functions” by “biochemical functions” in order to reflect this.

Line 50: “not only regulating the translation of the 50 canonical protein located downstream, but also producing short peptides (~25 AA) that can interact with it” While this is true for some cases -for instance, some examples given by Chen et al.-, uORFs can also produce proteins whose functions are independent of the main protein, so it would be good to highlight this possibility (e.g. PMID: 33406399 or PMID: 33449506).

This is correct, and a fair point, we have reworded that sentence to : “that can either interact with it¹⁴, or function independently^{15,16} ” to take into account this possibility, citing the articles suggested by the reviewer.

Line 108: Are transcribed lncRNAs the ones that had at least one ORF with RPKMRNA > 1?

In order to have a conservative analysis in light of possible alternative isoforms, our analysis is always ORF-centric, so a lncORF is considered to be transcribed if it has an RNA RPKM>1, however analyses concerning transcript expression e.g. those in Fig. S3 b-d, take into account the RNA RPKM of the whole transcript, as specified in those figure legends.

Line 154: Are there ORF length differences between the different ORF states? It would be interesting to see this in a plot.

This is a very interesting suggestion, see new Fig.S4g. Indeed translated lncORFs are twice as long as expected by chance (Couso and Patraquim 2017) and observed in ribo-bound-only lncORFs. We have added this result to section 5 concerning Phylogenetic conservation, as it supports that translated lncORFs are not a random group, but have been selected according to specific characteristics.

Line 156: I am a bit confused about what is a ‘translation event’. Please elaborate.

To clarify, we have re-worded the explanation in parenthesis: (framed RPFs that are either different in size, or come from different replicas or stages)

Line 167: IRIN should point to fig. 1G

Corrected.

Line 203: Are translated lncRNAs more highly expressed? For comparison, what % of protein-coding genes are translated? This is maybe described in the author's previous paper, but I miss this information in this manuscript.

Yes, we show that translated lncRNAs are more highly expressed (TE and Polysomal loading, RPKM RNA, etc.). See also point 4 of rev. 1 and point 3 of rev. 2; see fig. 4A on TE. We also specify that 90% of canonical ORFs are translated (added to text in line 192).

Line 322: The authors should show the distribution of ORF lengths per degree of conservation, separated by ORF stages.

Unfortunately, there are not enough lncORFs to do this, although as discussed above we can show the statistically significant difference between translated and not. If the reviewer's concern is that ORF length may impinge on conservation, this is fair and the main problem with BLAST and other engines which use cumulative scores. This is why we have developed GENOR, which goes beyond previous methods for short sequences (see Fig. 5b). Also, our Framing measurement with the binomial method is not length-dependent (Patraquim et al. 2020), so the fact that translation correlates with length is not an artifact.

Line 399: If ribosome profiling datasets of other tissues and conditions in *Drosophila melanogaster* are publicly available, it would be interesting to see how many of the 'ribo-bound ORFs' remain in the same state or if, otherwise, are highly translated at specific conditions or tissues.

We already do this analysis with the activated egg/unfertilized embryo data from *Kronja et al. 2014*. For S2 cells, we couldn't use the data from *Aspden et al. 2014* as it did not have reliable framing for lncORFs, but we obtained new samples with our improved protocol which are featured in this manuscript.

Line 413: "In turn, this purifying selection is strong proof that the encoded micropeptides are translated and functional." The ORFs were compared as a group so there is no individual evidence of possible selection and functionality. Hence, the sentence should refer to the enrichment in this group rather than considering that all the micropeptides are selected.

We have changed the wording to reflect this, it now reads: "In turn, the observed instances of purifying selection are strong proof that those encoded micropeptides are translated and functional, we also call the files with individual results together with the relevant figures, and the example in Fig. 5F.

Line 441: "However, unicellular eukaryote genomes include uORFs but not lncRNAs,". This statement is not correct. Unicellular eukaryote genomes also contain lncRNAs, although at lower numbers. Due to the small size of these genomes, most of these transcripts overlap protein-coding genes in antisense configurations. Since the authors are not only describing lincRNAs, antisense genes would be in principle

considered as lncRNAs. It would be good to know how many of the expressed and translated lncRNAs in *Drosophila* are antisense genes.

Please see Supplementary file 1, where the Number of antisense (as:CR) genes can be obtained: 30 translated lncRNAs (22% of the 136 translated lncRNAs) are annotated as asRNAs, but asRNAs represent 481 out of 2349 lncRNAs (20%), so asRNA translation doesn't seem to be enriched, when compared to other lncRNA types.

We have also reworded that statement as follows: " However, unicellular eukaryote genomes include uORFs, but to a lesser extent lncRNAs, and so the mechanisms for "gene birth" may have diversified during evolution."

Line 453: "Similarly, micropeptides produced by a lncRNA have been shown to enhance the function of miRNAs produced by the same lncRNA." How many lncRNAs which act as small RNA hosts are translated? Is there an enrichment of this RNA category in the set of translated RNAs?

We have investigated whether lncRNAs which undergo translation also produce miRNAs, the most common type of small RNA molecule hosted by lncRNAs (Fig. S4a). Interestingly, there seems to be mutual exclusion between lncRNA translation and miRNA production.

Line 641: What is the upper length cut-off? 150 or 450 codons? Did you see any lncRNA with translated ORFs above this length? Maybe a small % of lncRNAs that the authors don't find as translated contain longer ORFs, probably some unannotated protein-coding genes or pseudogenes.

There is no upper limit cut-off for well annotated *Dmel* lncRNAs, but we have now specified in methods that the upper-cut off for ORF identification in other species is 150 codons. Our new analysis with AnAblast (Fig. S1) shows that although translated lncORFs likely exclude pseudogenes, the genomic regions annotated as non-coding, likely including lncORFs identified as non-translated in this study, do include such sequences.

Line 643: This methods section is difficult to understand. Does this line mean that any internal ORF overlapping a longer one was removed? Does it refer to the same frame or any of the three possible frames? What are duplicates here? How was the set of 18,507 ORFs reduced to 16,335?

We have clarified this, this sentence now reads: "Finally, the coordinates of these ORFs were compared with other ORF classes (command *bedtools intersect -v -s -a*; no overlaps with both annotated coding sequences and uORFs) to obtain a set of 16,335 unique, non-overlapping lncORFs for further analysis."

Line 727: What is the minimum number of substitutions required to calculate dN/dS? The authors should make this information available in a table, including the total number of substitutions.

We have generated a new file (supplementary File 2) including the number and nature of the substitutions leading to the calculated dN/dS scores for each lncORF.

In order to obtain a dN/dS score a minimum of 2 substitutions are needed (1 non syn, 1 syn to avoid zeroes and infinities).

REVIEWER COMMENTS

Reviewer #1 (Remarks to the Author):

The authors have addressed some of my comments, and the manuscript has improved. The most relevant part of the manuscript remains the evolutionary analysis, and it still does not address the simple question – how many of the translated ORFs they identified in *D. melanogaster* are conserved *above what is expected by chance*, which would make them likely to encode functional peptides. Tools like PhyloCSF can address this question, but in this case they appear to report only a small minority of ORFs as significant, for reasons that the authors do not elaborate on the manuscript. The authors show that the dN/dS ratio in the ORFs *as a group* are higher than in ribosome-associated ORFs, but since in the latter group the frame is much more difficult to determine, it is not clear what this result really means (as the authors compare frame-confirmed ORFs with ORFs in which the frame, as far as I understand, is generally ambiguous). They also show some difference in the GENOR conservation scores they came up with, but since these are very ad hoc scores with no statistical interpretation attached to them, it is also not clear what this difference means (or if it is statistically significant). The question of “signal-background estimation” should be possible to address – the authors can e.g. take random regions in the same lncRNAs, with similar GENOR score, and then ask in how many lncRNAs, the dN/dS score associated with the actual translated ORF is better (more indicative of peptide sequence conservation) than expected by chance. This will allow them to come up with signal-noise estimation, very useful estimations for how many of the ORFs they report as translated are likely to be under evolutionary pressure to encode peptides. It seems that the authors can also deduce the signal-background estimate from their representation factor calculation. If this number is high (i.e., many ORFs translated and not expected by chance), it strengthens their argument. If it is lower, it weakens it. In any case, these numbers are very important for the interpretation by the community, more than the overall trend for conservation when all the ORFs are considered together, and so I believe this analysis can substantially strengthen the manuscript.

Reviewer #2 (Remarks to the Author):

The authors successfully addressed only some of my comments, mostly minor. However, I empathise with the authors' explanations for why they were not able to address these comments.

I am somewhat disappointed that the authors were not able to come up with better terminology than “non-canonical” but I agree that the issue of terminology is very hard to solve and perhaps this should be done elsewhere. Besides, the authors did make some noticeable terminological improvements including in the title.

While RNA 5' ends mapping is highly desirable, it is indeed could be difficult to achieve and may not substantially alter the conclusions. I get the authors' point that the lower quality of 5' end mapping argues for the robustness of the identified patterns, as they are being seen above higher noise.

The authors did not address my question about the initiating ribosome load at different RNAs classes, but I agree that this would be somewhat a new research direction and didn't have high expectations that the authors would do that. This also may not be easy without addressing the previous and the following issues.

The authors also did not extend their analysis to include non-AUG initiation. I don't think it is necessary, but here I disagree with the authors' argumentation for why not to do it. Based on the data currently available for mammalian cells, non-AUG translation seems highly pervasive, just much less efficient than AUG initiation on average. So, I think it is neither controversial nor minor (in terms of frequency). I suspect that the situation may be similar in *Drosophila*. However, looking at low efficient translation at lowly abundant RNAs could be difficult and may complicate the analysis and reduce the confidence in the resulting pool of all detected translated ORFs. Thus, I do not insist that this should

be done. However, I suggest bringing this issue up in the manuscript. I strongly suspect that the lower triplet periodicity in Fig. S1 for all IncORFs is due to competing translation from non-AUG initiators at the overlapping ORFs.

Panel D in Fig. S2 is missing.

Reviewer #3 (Remarks to the Author):

The authors correctly addressed all my points. I do not have further comments and I would recommend the article for publication.

I only have a couple of minor suggestions which I think would be important to add to the manuscript:

-Micropeptide translation: The authors assume that all ORFs translated from lncRNAs encode for microproteins, although there is still some controversy on this topic, e.g. most of these novel ORFs do not have evidence in mass-spectrometry. My personal suggestion is that the authors should use the discussion to highlight the need for additional future work to validate the translation of stable micropeptides from ORFs in lncRNAs.

-Cut-off of 2 substitutions to obtain dN/dS: The authors could show the results from fig. 5E remain significant when higher cut-offs on the number of substitutions are added (5?). This is an important result for the manuscript and it would make clear that using a low cut-off does not affect the observed differences. I realized panel 5D is not referenced in the text.

RESPONSE TO REVIEWERS

Second Revision, Nature Communications

Response to Reviewer #1:

We agree with the reviewer's main comment that signal-background estimation is important to interpret the results pointing at a higher level of conservation in translated ORFs as a group - a key result of our work. We now address the question of how many of the translated lncORFs identified in *D. melanogaster* are conserved above what is expected by chance, assessing the statistical significance of the overlap between translated and conserved lncORFs. As suggested, we calculated the overrepresentation factors for Dsim and Dsec, species where we best observe this correlation. We obtained a high overrepresentation of the overlap between robustly translated and conserved (GENOR >50) lncORFs for both species, which is highly significant ($p < 0.00001$). These results statistically support the observation that translated lncORFs show clear phylogenetic signal consistent with conservation in closely-related species of the *Drosophila* genus; They have been added to a new panel (c) in *Supplementary Figure 5* (with a corresponding legend and reference to the panel in the main results text, **Line: 365**, section 5. *Phylogenetic conservation supports lncRNA translation and micropeptide function*, 3rd paragraph). We have also added an excerpt of Table S1 in panel 5S d, highlighting some individual examples of conserved lncORFs in Dsim, hoping that this will further encourage readers to access File S1, which displays expression, translation and conservation data for all lncORFs.

As pointed out before, we show statistically significant differences in conservation and dN/dS scores between lncORFs differing by their translation status. These sequences are otherwise subjected to the same constraints and background "noise" is as such always taken into account when reporting these differences (such as the plots in Fig 5 e, showing significantly different medians and distribution of the dN/dS values of these two classes of lncORFs). Here it is important to add a clarification relating to the question of frame determination in Ribo-Bound lncORFs: all lncORFs were predicted using getORF, a tool that finds overlapping ORFs in different frames. The lncORFs are predicted before framing is analysed. As such, it is true that lncORFs are considered translated if the reads fit the frame, but all lncRNA frames may contain lncORFs and reads. This renders the observation of translation wholly independent from frame determination.

We thank this reviewer's comments for helping us highlight the significance of our findings on lncORF conservation.

Response to Reviewer #2:

We are grateful that the reviewer agrees with our reasoning for not carrying out some of his proposed lines of research. Regarding the issue of non-AUG initiation, the reviewer is correct the low efficient translation of non-AUG sites at already lowly abundant RNAs would complicate the analysis and reduce the confidence in the resulting pool of all detected translated ORFs. However, we agree that it would be beneficial to discuss this ORF group. We have therefore added the following paragraph in the discussion (**line 453, 4th paragraph**): "Although a role for non-AUG translation has been suggested in shaping translation programs, we have not addressed it in the present study, as its low efficiency, when compared to canonical START codons, is expected to significantly hamper its detection in the case of lncRNAs. It is however conceivable that some of the ribo-bound or limited lncORFs are in fact translated but

do not show framing due to leaky ribosomal occupation signal coming from an overlapping non-AUG lncORFs (Adreev et al. Gen Biol 2022).”

We see a Panel S2d in our submission, and assume this to be a compatibility/printing issue; to preclude it, we have replaced the digital version of the panel for a flattened .pdf object. A reference to this panel, which was absent and partially addresses the relationship between ribosomal load and productive translation, has been added in the results section, under 2. *Developmental regulation of lncORF* (Line 194: “Further, we observe that growing amounts of ribosomal association do not necessarily dictate productive translation, as indicated by framing (Fig. S2d) suggesting that the amount of ribosomal association is not the only factor that favors lncORF translation, which supports the importance of the quality of ribosomal engagement.”). (see also response to reviewer 3).

Response to Reviewer #3:

We thank the reviewer’s comments, which have improved the manuscript, and share the reviewer’s main comment that translation of lncNRAs (including robust translation) could have functional consequences besides the putative micropeptide encoded (e.g. the translation of the ORF itself is functional, with the peptide as free to evolve in sequence, as with some uORFs). We had already mentioned this in the manuscript discussion (lines 501-509 of previously submitted text, now lines 516-254), and have now added a sentence in this paragraph (under *function of lncRNA translation*) clarifying that we do not expect *all* robustly translated lncORFs to exert their function at the protein level (Lines 519-521 “We find that robustly translated lncORFs as a group contain clear evidence for the conservation of AA sequences, an indication that a good proportion of this cohort is exerting its function at the protein level.”). However, we see the marks of natural selection on amino-acid composition for the robustly translated lncORFs as a group; as stated in the discussion, we believe that this conservation of amino acid composition is difficult to explain without a (however limited) functional requirement for the micropeptide sequence itself. We do not believe that all lncORFs in this group encode strongly constrained amino acid sequences, as with canonical proteins. Rather, we are suggesting that perhaps not all but some robustly translated lncORFs are novel genes, with limited but clear signals for the action of natural selection shifting from nucleotide to amino acid sequence within the *Drosophila* genus - and wish to emphasize this in our article.

Mass-spectrometry has been shown to be lacking for translated and functional micropeptides, as its detection threshold is very high for short AA sequences (Patraquim *et al.* Gen Bio, 2020; Bazzini *et al.* 2014, EMBO journal;) and, also in line with other colleagues, we would not use it as single criterium in the validation of RiboSeq data (Mudge *et al.*, 2022, Nat Biotechnology). We also share the belief that additional work to validate lncORF translation is needed (as mentioned in the discussion, line 550), and will require experimental genetic engineering of individual nucleotide sequences within the ORF, such as we produce in this work.

The idea of progressively restrictive substitutions cutoffs for dN/dS was developed for longer, canonical sequences which have a mean a mean of 500 AA (± 50) – five to ten times the amount of information available to us. For our set of very short sequences, with between 12 and 30 AA in most cases, a threshold of 5 substitutions will mean up to 50% of the coding sequence accumulating mutations in a very short evolutionary timescale. In fact, only 19% of smORFs have 5 substitutions or more in Dsim (File S2), including the example shown in Fig. 5f, which has 7 substitutions in 48 codons. We consider that we are at the appropriate level of sensitivity

for our set of very short sequences, with between 12 and 30 AA in most cases, and believe that higher arbitrary cutoffs, would restrict analysis to a few lncORFs, and although more stringent, this would limit our understanding of the evolutionary dynamics in this ORF class.

We assume the reviewer refers to panel Sup.2D; we have added a reference to this panel at an appropriate point in the results section, under *2. Developmental regulation of lncORF translation*.

REVIEWERS' COMMENTS

Reviewer #1 (Remarks to the Author):

The authors have addressed my comments in a satisfactory manner, and I believe the manuscript has improved as a result.

A minor comment remains: In Supp. Figure 5C, the authors should show the % each group represents of the total ORFs considered, rather than of the ORFs in either group, this will make it clearer how they get to the 8-fold and 4-fold over-representation in the overlap over expected by chance.

Reviewer #3 (Remarks to the Author):

Thanks to the authors for clarifying my two minor suggestions. I don't have any further comments.